# Suppressing electron-phonon coupling in organic photovoltaics for high-efficiency power conversion

Yuanyuan Jiang[1,2,4], Yixin Li[1,4], Feng Liu[1,4], Wenxuan Wang[1,2], Wenli Su[3], Wuyue Liu[1], Songjun Liu[1,2], Wenkai Zhang[3], Jianhui Hou[1,2], Shengjie Xu[1]✉, Yuanping Yi[1]✉ & Xiaozhang Zhu[1,2]✉

The nonradiative energy loss ($\Delta E_{nr}$) is a critical factor to limit the efficiency of organic solar cells. Generally, strong electron-phonon coupling induced by molecular motion generates fast nonradiative decay and causes high $\Delta E_{nr}$. How to restrict molecular motion and achieve a low $\Delta E_{nr}$ is a sticking point. Herein, the free volume ratio (FVR) is proposed as an indicator to evaluate molecular motion, providing new molecular design rationale to suppress nonradiative decay. Theoretical and experimental results indicate proper proliferation of alkyl side-chain can decrease FVR and restrict molecular motion, leading to reduced electron-phonon coupling while maintaining ideal nanomorphology. The reduced FVR and favorable morphology are simultaneously obtained in AQx-6 with pinpoint alkyl chain proliferation, achieving a high PCE of 18.6% with optimized $V_{OC}$, $J_{SC}$ and FF. Our study discovered aggregation-state regulation is of great importance to the reduction of electron-phonon coupling, which paves the way to high-efficiency OSCs.

Organic solar cells (OSCs) as a promising photovoltaic technology have been drawing great attention, which presents great potential in flexibility, large-area fabrication, semitransparency and etc[1-13]. Great progresses have been made in power conversion efficiencies (PCEs) recently, benefitting from subtle molecular design[14-27] and device engineering[28-39]. According to the Shockley-Queisser (SQ) theory, there should be only inevitable radiative energy loss above bandgap in an ideal solar cell[40,41], however, a considerable nonradiative energy loss ($\Delta E_{nr}$) below bandgap presents in realistic OSCs, which has been recognized as a key issue for higher PCE[42,43]. Essentially, the energy loss below bandgap originates from the relaxation processes from the excited states (local excited states of individual materials and intermolecular charge-transfer states at donor/acceptor interface) to the ground states accompanied with photon emission and phonon transmission, corresponding to radiative energy loss ($\Delta E_r$) and $\Delta E_{nr}$, respectively[44-47]. The competitive relationship between the nonradiative and radiative relaxation processes suggests suppressed nonradiative decay rate ($k_{nr}$) is conducive to improving photoluminescence quantum yield (PLQY) by the equation of PLQY = $\frac{k_r}{k_r + k_{nr}}$[48,49]. According to the energy-gap law, nonradiative decay rate $k_{nr}$ can be described as $k_{nr} \propto e^{-\frac{2\pi\gamma\Delta E}{\hbar\omega_M}}$[50], where $\omega_M$ is the maximum and dominant vibrational frequency and $\Delta E$ is the excitation energy, indicating that the lower bandgap, the higher nonradiative decay rate and lower PLQY. Recently, Brades et al. confirmed that the higher PLQY of the active-layer material featuring a lower optical bandgap determines the lower limit of $\Delta E_{nr}$[41]. Therefore, the following question on the way to efficient OSCs with low $\Delta E_{nr}$ is how to restrict molecular motions or in other words, suppress electron–phonon coupling, especially in the lower bandgap organic active-layer materials[49,51].

Typically, organic semiconductor materials consist of a conjugated backbone and flexible side chains, in which the backbone

[1]Beijing National Laboratory for Molecular Sciences, CAS Key Laboratory of Organic Solids, Institute of Chemistry, Chinese Academy of Sciences, Beijing 100190, China. [2]School of Chemical Sciences, University of Chinese Academy of Sciences, Beijing 100049, China. [3]Department of Physics and Applied Optics, Beijing Area Major Laboratory Center for Advanced Quantum Studies, Beijing Normal University, Beijing 100875, China. [4]These authors contributed equally: Yuanyuan Jiang, Yixin Li, Feng Liu. ✉e-mail: xushengjie@iccas.ac.cn; ypyi@iccas.ac.cn; xzzhu@iccas.ac.cn

forms π–π stacking for charge carrier transport, and side chains can not only bring solution processability but also significantly affect the aggregation pattern in solid state[52,53]. The current state-of-the-art nonfullerene acceptors (NFAs) exhibit a non-linear configuration[54], such as Y series with a banana geometry presenting various packing modes, which forms multiple charge transport channels and delivers superior charge transport property[24,55]. However, these rigid fused backbones inevitably produce voids, named as free volume, between adjacent backbones in aggregation state, which generates the probability for molecular thermal motion, and thus endows NFAs with lower PLQY[56]. Vandewal et al. demonstrated that intrinsic high-frequency carbon–carbon stretch vibrations are a dominant cause for the nonradiative decay process, a major source for $\Delta E_{nr}$ in fullerene-based OSCs[57,58]. In addition to these high-frequency stretch vibrations, the $\Delta E_{nr}$ of NFAs that features an extended (quasi-) planar π-conjugated backbone also is related to out-of-plane bending and in-plane torsion. Since the microscopic molecular thermal motions are difficult to analyze quantitatively, we assume that the free volume ratio (FVR) may be an indicator, allowing a straightforward platform to investigate how to reduce $\Delta E_{nr}$ via rational molecular design. Aggregation-induced emission (AIE) mechanism indicates that restriction of intramolecular motions in solid state can slow down the $k_{nr}$ but have little influences on the $k_r$[59], leading to higher PLQY. Inspired by the AIE science, we expect that flexible alkyl side chains can act as "glue" by lipsoluble supramolecular assembly[60,61], and decrease the FVR for restricted molecular motions, leading to an improved PLQY in NFAs and a lower $\Delta E_{nr}$ in OSCs.

Our group have been devoted to developing high-efficiency NFAs, and AQx-type acceptors featuring a quinoidal-enhancing quinoxaline moiety were designed and synthesized in 2019[62], achieving a record certified PCE when it was published[63,64]. Recently, Wei et al. demonstrated that AQx-type acceptors exhibit much lower reorganization energy than Y6[65], leading to better charge transport property, longer exciton diffusion length, reduced $\Delta E_{loss}$ and thus allowing superior potential to achieve higher PCEs. Herein, we performed a precise alkyl side-chain modulation on the AQx-type acceptors, which provides a feasible and convenient approach to decrease $\Delta E_{nr}$. In contrast to the most research focus on the symmetrical side-chain modulations[22,24,66,67], we selectively proliferated alkyl side chain. Therefore, AQx-2 with symmetrical 2-ethylhexyl, 2,2′-((2Z,2′Z)-((13-(2-ethylhexyl)-14-(2-hexyldecyl)-3,10-diundecyl-13,14-dihydrothieno[2″,3″:4′,5′]thieno[2′,3′:4,5]pyrrolo[3,2-f]thieno-[2″,3″:4′,5′]thieno[2′,3′:4,5]pyrolo-[2,3-h]quinoxaline-2,11-diyl)bis(methaneylylidene))bis(5,6-difluoro-3-oxo-2,3-dihydro-1H-indene-2,1-di-ylidene))dimalononitrile (AQx-6) with asymmetrical 2-ethylhexyl and 2-hexyldecyl and 2,2′-((2Z,2′Z)-((13,14-bis(2-hexyldecyl)-3,10-diundecyl-13,14-dihydrothieno[2″,3″:4′,5′]thieno[2′,3′:4,5]pyrrolo[3,2-f]-thieno[2″,3″:4′,5′]thieno[2′,3′:4,5]pyrrolo[2,3-h]quinoxaline-2,11-diyl)bis(methaneylylidene))bis(5,6-di-fluoro-3-oxo-2,3-dihydro-1H-indene-2,1-diylidene))dimalononitrile (AQx-8) with symmetrical 2-hexyldecyl were designed and synthesized, with regulated alkyl side chains attached on the nitrogen atoms of the pyrrole rings. The theoretical and experimental results demonstrated that the prolonged alkyl side-chain helps to decrease the FVR in AQx-6 and AQx-8, which is beneficial to reduction the electron–phonon coupling, leading to enhanced emission efficiency and reduced $\Delta E_{nr}$. However, nanomorphology that includes phase separation and crystallinity will be affected by the prolonged alkyl side-chain length, making differences in exciton separation, charge transport and recombination processes. Notably, AQx-6-based OSC achieved the subtle balance between suppressed electron–phonon coupling for low $\Delta E_{nr}$ and efficient exciton dissociation as well as charge transport for high $J_{SC}$ and FF. Eventually, the AQx-6-based binary OSC delivers a remarkable PCE of 18.6% (certified PCE: 18.4%) with a $V_{OC}$ of 0.892 V, $J_{SC}$ of 26.8 mA cm$^{-2}$ and FF of 77.8%.

## Results

### Molecular design and theoretical calculations of AQx-type acceptors

Based on the highly performed NFA AQx-2, AQx-6 and AQx-8 were designed by gradually proliferating 2-ethylhexyl substituents attached on the sp$^2$-hybridized nitrogen atoms in the pyrrole rings with 2-hexyldecyl substituents, the molecular structures of AQx-2, AQx-6 and AQx-8 are shown in Fig. 1a, and the relevant synthetic routes were shown in Supplementary Fig. 1. In order to explore the effect of varied alkyl side chains on molecular motion, the molecular packing structures and FVRs of three acceptors have been investigated by atomistic molecular dynamics simulations (the simulation details see Supplementary information). As shown in Fig. 1b–d, AQx-2 with the shorter alkyl side-chain possesses a FVR of 38.35%, and longer alkyl side chain in AQx-6 and AQx-8 allow decreased FVR values of 37.87% and 37.95%, respectively, which is consistent with what we expected. The reduced FVR allows less space for molecular motion, leading to suppressed nonradiative decay process. The PLQY and time-resolved photoluminescence (TRPL) measurements (Supplementary Fig. 8) were carried out to investigate emission efficiency and exciton lifetime ($\tau$) in three acceptors. As shown in Fig. 1e and Supplementary Table 1, with the increased alkyl side-chain length, PLQY gradually increased from 0.39% (AQx-2) to 0.67% (AQx-6), and then to 1.54% (AQx-8), the higher PLQY contributes to a lower $\Delta E_{nr}$ in relevant OSCs[68]. The $\tau$ values for AQx-2, AQx-6 and AQx-8 are 60.99, 104.07 and 103.45 ps, respectively. Based on the equation of PLQY = $\frac{k_r}{k_r + k_{nr}} = \tau k_r$, the $k_r$ and $k_{nr}$ are 6.40 × 10$^7$ and 1.63 × 10$^{10}$, 6.44 × 10$^7$ and 9.55 × 10$^9$, 1.49 × 10$^8$ and 9.52 × 10$^9$ s$^{-1}$ for AQx-2, AQx-6 and AQx-8, respectively. Compared to AQx-2, the $k_{nr}$s get the reduction of an order of magnitude in AQx-6 and AQx-8, which is well consistent with the variation of the FVR. The faster radiative decay process and higher PLQY are observed in AQx-8, which can be attributed to a slightly looser π–π stacking that is caused by long alkyl side-chain, leading to a reduced H-aggregation[69]. The similar $k_{nr}$s in AQx-6 and AQx-8 agree well with their similar FVRs. Thus, we can conclude that prolonged alkyl side chain helps to suppress nonradiative decay process associated to molecular motion via "adhesive" effect, which is of great importance to reduce $E_{loss}$ and ultimately realize high performance. Such a conclusion can be confirmed by the experimental $E_{loss}$ of AQx-based systems and the detailed analysis will be discussed below.

### Photoelectric property

The molecular geometry without alkyl side chain in a single crystal of AQx-2 is exhibited in Fig. 2a and the single crystal was successfully developed through diffusing diethyl ether vapor into chloroform solution. AQx-2 presents a banana-shaped geometry, and such a configuration is inevitable to form free volumes between rigid backbones. As shown in Fig. 2c, the molecular packing in single crystal of AQx-2 presents a dense wave-like molecular arrangement with hexagon-shaped voids. For each molecule, it gives four overlaps with adjacent molecules, in which one type is J-aggregation between the end group and the other is H-aggregation between backbones, with the π–π stacking distances of 3.341 and 3.424 Å (Fig. 2b)[70]. The compact 2D brick-layer arrangement allows efficient charge transport, contributing to less charge combination and desired FF for high performance in device. The UV–Vis absorption spectra of acceptors in chloroform solution and thin film and donor film are shown in Fig. 2d, in which D18[23] as a donor exhibits a complementary absorption with AQx-type acceptors for more photon harvesting. With the similar molecular structure, three acceptors present nearly identical absorption with a maximum absorption peak at 732 nm in diluted chloroform solution, and their absorption spectra in thin film show bathochromic-shift absorption, suggesting strengthened and ordered aggregation in solid state[70]. The AQx-2, AQx-6 and AQx-8 show maximum absorption peaks

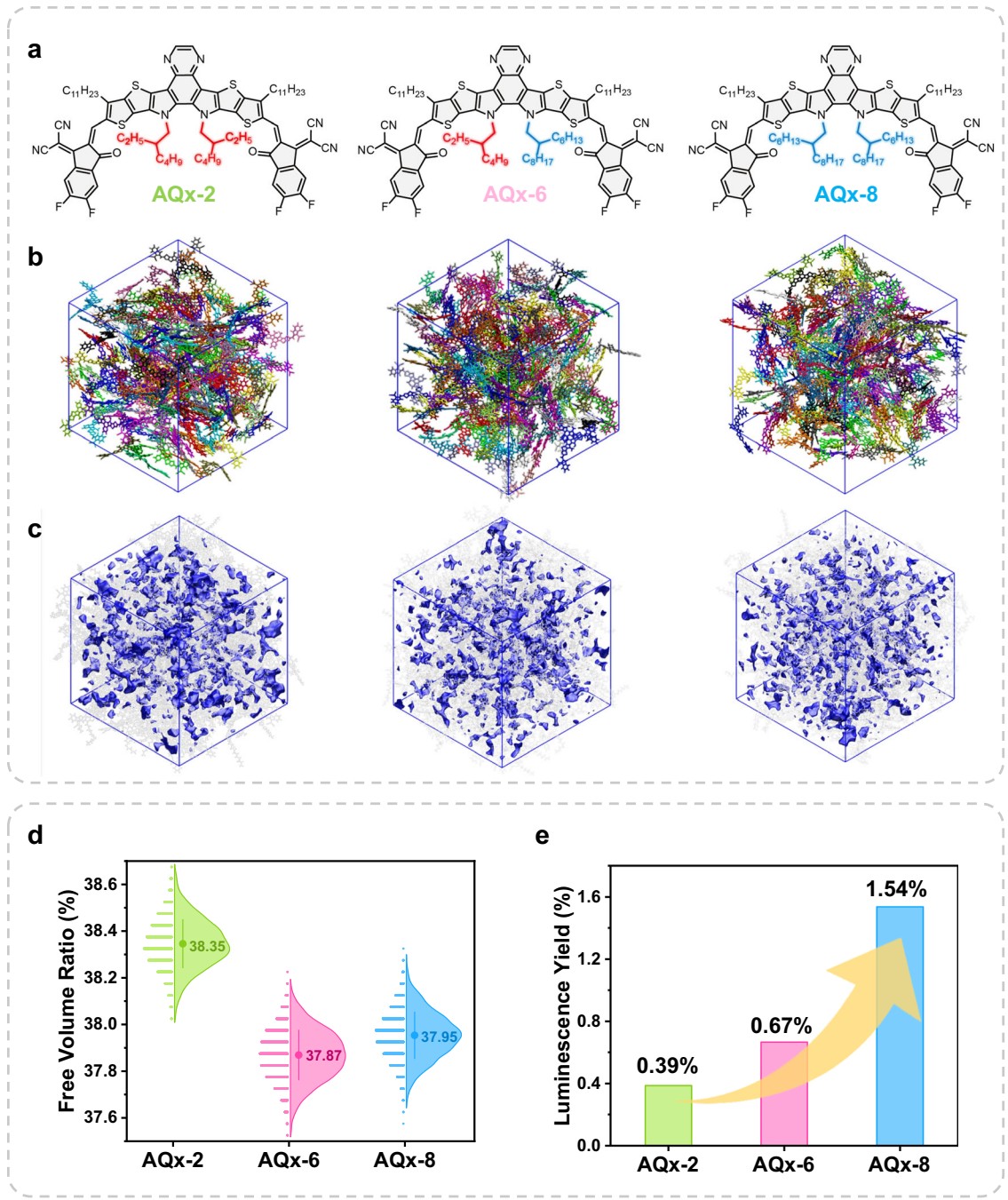

**Fig. 1 | Chemical structures and theoretical calculation. a** Chemical structures. **b** Representative snapshots of MD simulated molecular packing morphologies. **c** Illustration of free volumes. **d** Free volume ratio. **e** Luminescence yields of AQx−2, AQx-6 and AQx-8 films.

at 825, 815 and 812 nm with absorption onset at 941, 936 and 933 nm, corresponding to optical gap of 1.32, 1.32 and 1.33 eV, respectively. The blue-shifted absorption from AQx-2 (825 nm) to AQx-8 (812 nm) can be ascribed to gradually prolonged alkyl side, which induces the reduced $\pi$−$\pi$ stacking in AQx-8 pristine film. D18 film shows the absorption range of 400−650 nm, which is complementary to those of the AQx-type acceptors in a large part of the solar spectrum, benefiting solar-spectrum utilization. The cyclic voltammetry (CV) measurements were performed to measure the energy levels (Supplementary Fig. 11) that includes the highest occupied molecular orbital (HOMO) and the lowest unoccupied molecular orbital (LUMO) energy levels of three acceptors. As shown in Fig. 2e, the HOMO energy levels of AQx-2, AQx-6 and AQx-8 are −5.62, −5.61 and −5.60 eV and the LUMO energy levels are −3.88,

−3.86 and −3.85 eV, respectively. There is no obvious difference of HOMO and LUMO energy levels among three acceptors, suggesting fine-tune of alkyl side chain has little impact on energy levels, which is beneficial to investigate the relationship among alkyl side-chain substitution, aggregation structure and nonradiative decay path associated with the electron−phonon coupling. Moreover, as shown in Supplementary Fig. 12, AQx-type acceptors possess good thermal stability (5% weight loss at 345, 347 and 345 °C for AQx-2, AQx-6 and AQx-8, respectively) according to the thermogravimetric analysis (TGA).

### Photovoltaic performance
All OSCs based on the conventional device configuration of glass/ITO/PEDOT:PSS/D18:acceptor/PDINN/Ag, in which ITO is indium tin oxide,

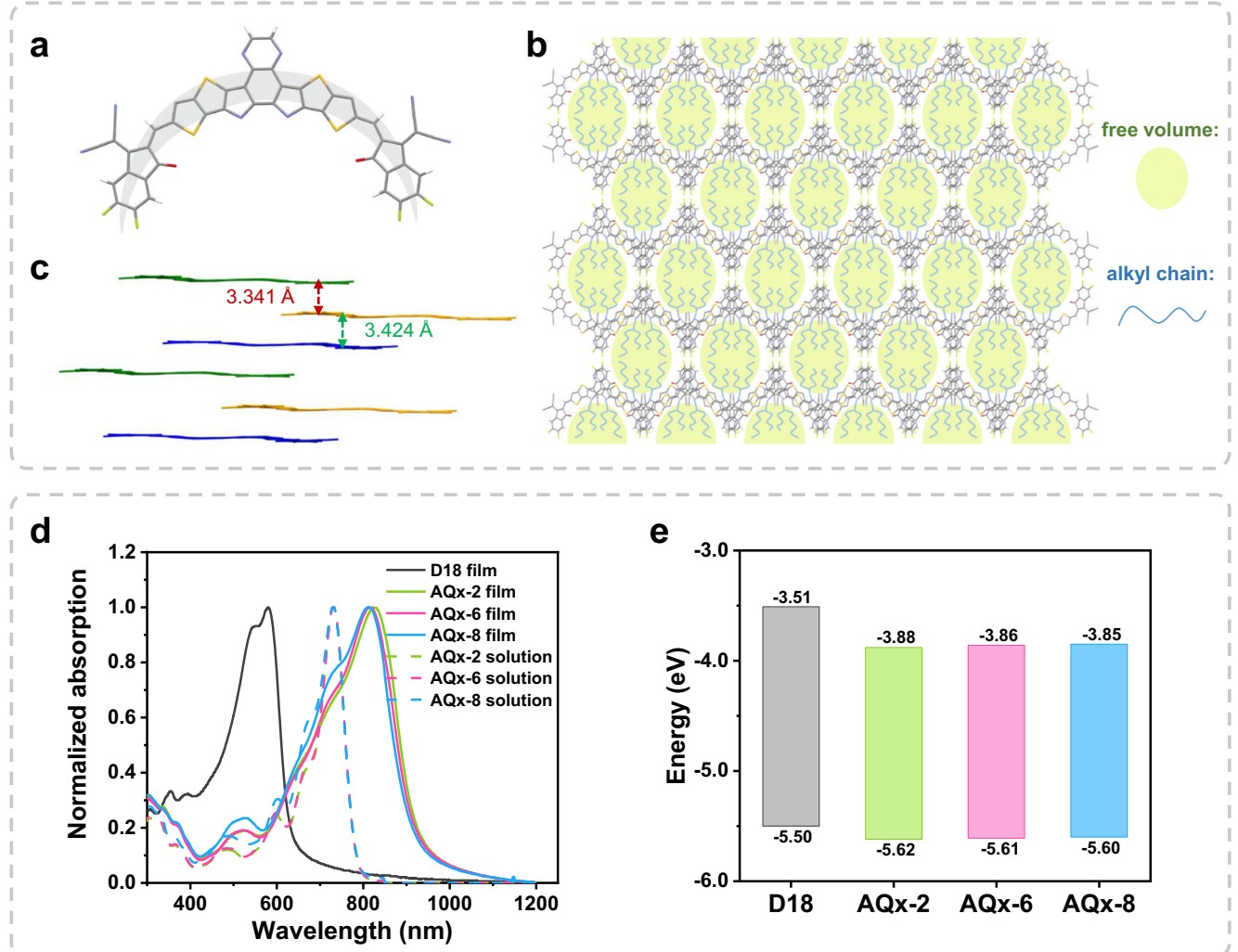

**Fig. 2 | Single-crystal structure and photoelectric properties of AQx-type NFAs.**
**a** The single-crystal molecular configuration of AQx−2 without alkyl chain for clarity
(CCDC number: 2161927). **b** π−π stacking distance. **c** Molecular packing patterns of
AQx-2 in single crystal (The inserted alkyl side chain is schematic diagram). **d** UV–Vis-NIR spectra. **e** Energy levels diagram.

PEDOT:PSS is poly(3,4-ethylenedioxythiophene):poly(styrenesulfonate) that is used as hole transport layer, PDINN acts as electron transport layer[71]. Supplementary Tables 2–7 are summarized the detailed device optimization process including the donor:acceptor ratio and thermal annealing temperature. Table 1 gives the detailed photovoltaic parameters of three devices and Fig. 3a shows the corresponding current density-voltage (J-V) curves. The devices based on D18:AQx-2 and D18:AQx-8 show passable PCE of 17.2 and 17.1% with $V_{OC}$ of 0.868 and 0.913 V, $J_{SC}$ of 26.1 and 24.7 mA cm$^{-2}$, FF of 76.0 and 75.7%, respectively. The AQx-2-based device exhibits the higher $J_{SC}$ (26.1 mA cm$^{-2}$), while the higher $V_{OC}$ (0.913 V) is observed in AQx-8-based device. As for the AQx-6 with asymmetrical alkyl side chain, the corresponding cells afford an improved $V_{OC}$ of 0.892 V and the highest $J_{sc}$ of 26.8 mA cm$^{-2}$ as well as FF of 77.8%, thus a champion efficiency of 18.6% was obtained, which is one of the highest PCE among the reported bulk-heterojunction binary OSCs (Supplementary Fig. 13). Besides, the certification of D18:AQx-6 performance was carried out in National Institute of Metrology (NIM), China, and a certified PCE of 18.4% with a $V_{OC}$ of 0.874 V, a $J_{SC}$ of 26.7 mA cm$^{-2}$ and an FF of 79.1% was obtained (Supplementary Fig. 14). The external quantum efficiency (EQE) curves are shown in Fig. 3b, and the integrated $J_{SC}$s from EQE curves are consistent with the $J_{SC}$ values from J−V measurements within 5% error. Noted that despite three acceptors give different absorption edge, AQx-2-, AQx-6- and AQx-8-

based devices present almost overlapped EQE curve edges, 944, 941 and 937 nm, which corresponds to the bandgap ($E_g$) of 1.314, 1.317 and 1.323 eV, respectively. According to the equation of $E_{loss} = E_g − qV_{oc}$, where $q$ is element charge, the $E_{loss}$s are 0.446 eV for D18:AQx-2, 0.425 eV for D18:AQx-6 and 0.410 eV for D18:AQx-8, indicating the prolonged alkyl chain can lower the $E_{loss}$ and elevate the value of $V_{OC}$ (Fig. 3c). Compared to AQx-2-based OSC, AQx-6-based OSC displays elevated $V_{OC}$ value and obtains highest $J_{SC}$ and FF, which produces the best performance among three devices.

The exciton generation, dissociation and charge recombination processes were performed to better understanding performance difference in three devices. The dependence of photocurrent density ($J_{ph}$) on the effective voltage ($V_{eff}$) was performed to explore exciton dissociation and charge extraction probability. The $J_{ph}$ is equated to $J_{light} − J_{dark}$, where $J_{light}$ and $J_{dark}$ is current density on illumination and dark condition, respectively. The $V_{eff}$ is defined as $V_0 − V_{app}$, where $V_0$ is the voltage when $J_{ph} = 0$ and $V_{app}$ is the applied voltage. The exciton dissociation probability $P(E,T)$ can be calculated with $J_{ph}/J_{sat}$ (Supplementary Fig. 15). Under the maximum power out condition, $P(E,T)$s are 98.7, 99.4 and 98.2% for AQx-2-, AQx-6- and AQx-8-based cells and under the short-circuit condition, $P(E,T)$s were 87.2 90.3 and 85.8% for AQx-2-, AQx-6- and AQx-8-based cells, indicating superior exciton dissociation and charge extraction process existed in AQx-6-based

**Table 1 | Photovoltaic parameters of D18:AQx-2, D18:AQx-6 and D18:AQx-8 based OSCs under the illumination of AM 1.5 G, 100 mW cm$^{-2}$**

| Acceptors | $V_{oc}$ (V) | $J_{SC}$ (mA cm$^{-2}$) | $J_{SC}^{cal,b}$ (mA cm$^{-2}$) | FF (%) | PCE (%) | $\Delta E_{nr}^{c}$ (eV) | $\Delta E_{loss}^{d}$ (eV) |
|---|---|---|---|---|---|---|---|
| AQx-2[a] | 0.868 (0.869 ± 0.003) | 26.1 (25.8 ± 0.2) | 25.4 | 76.0 (75.6 ± 0.3) | 17.2 (17.0 ± 0.11) | 0.206 | 0.529 |
| AQx-6[a] | 0.892 (0.890 ± 0.002) | 26.8 (26.7 ± 0.1) | 25.5 | 77.8 (77.1 ± 0.5) | 18.6 (18.3 ± 0.10) | 0.198 | 0.518 |
| AQx-8[a] | 0.913 (0.909 ± 0.003) | 24.7 (24.9 ± 0.1) | 23.9 | 75.7 (75.0 ± 0.4) | 17.1 (17.0 ± 0.05) | 0.178 | 0.493 |
| AQx-6[e] | 0.874 | 26.7 | – | 79.1 | 18.4 | – | – |

[a]Average values with standard deviation were obtained from 10 individual devices.
[b]The calculated $J_{SC}$ values from the EQE curve.
[c]Calculated using eq. $\Delta E_{nr} = -(kT/q)\ln(EQE_{EL})$.
[d]Calculated using eq. $\Delta E_{loss} = E_g - qV_{OC}$, in which $E_g$ is determined the crossing point of absorption and emission.
[e]The certified PCE obtained from National Institute of Metrology (NIM), Chain.

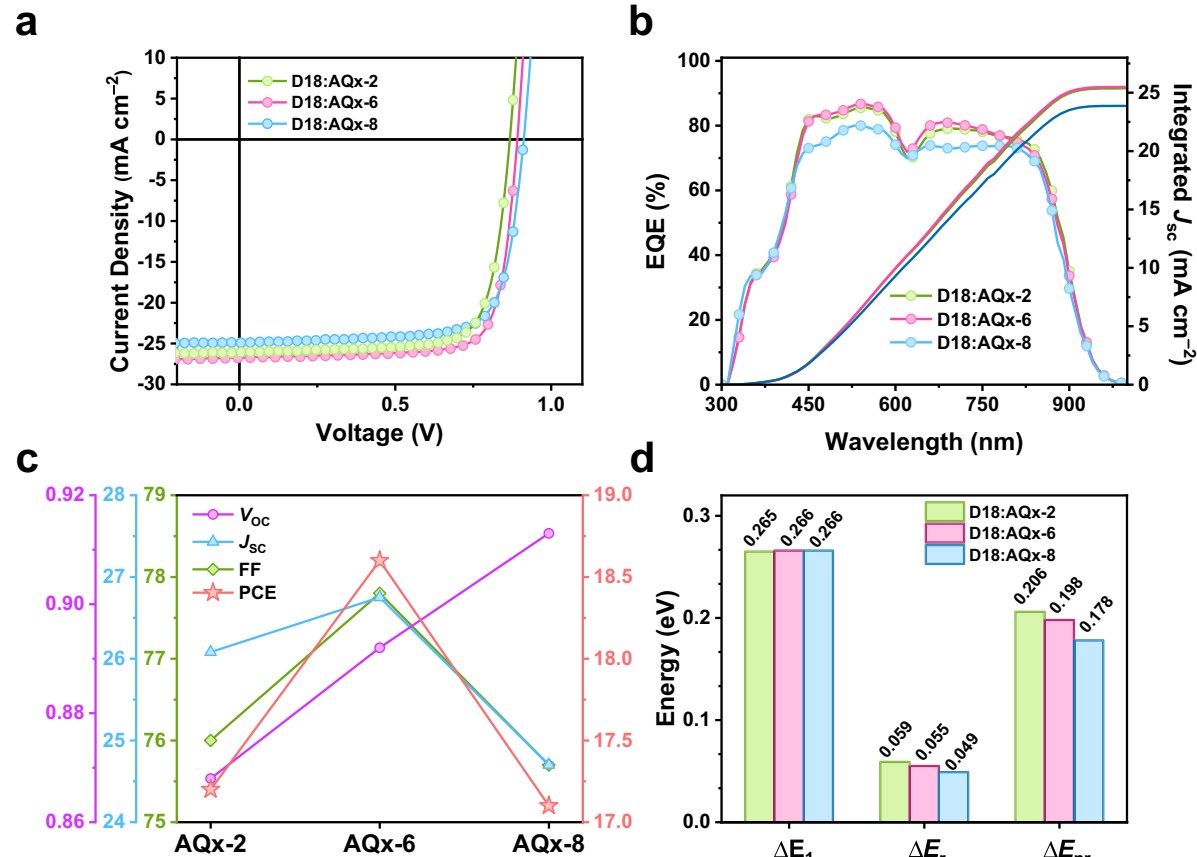

**Fig. 3 | Photovoltaic performance of OSCs based on AQx-type NFAs.**
**a** Characteristic J–V curves. **b** EQE curves. **c** Photovoltaic parameters variation for AQX-based OSCs ($V_{OC}$: V; $J_{SC}$: mA cm$^{-2}$; FF: %; PCE: %.). **d** Energy loss: $\Delta E_1$, $\Delta E_r$ and $\Delta E_{nr}$.

binary cell. In addition, charge recombination behavior in the three OSCs was examined by the dependence of $V_{OC}$ and $J_{SC}$ on light density ($P_{light}$). The relationship between $V_{OC}$ and $P_{light}$ can be described as $V_{OC} \propto (nkT/q)\ln P_{light}$, where $k$, $T$, $q$ are the Boltzmann constant, absolute temperature and elemental charge, respectively. When $n$ is close to 1, the bimolecular recombination is the dominated mechanism in devices. As displayed in Supplementary Fig. 16, the slope values for AQx-2-, AQx-6- and AQx-8-based devices are 1.05, 1.01 and 1.09 $kT/q$, respectively, suggesting all devices are subject to bimolecular recombination. The lowest slope value in AQx-6-based OSC means the trap-assisted recombination was well-mitigated. The relationship between $J_{SC}$ and $P_{light}$ obeys the law of $J_{SC} \propto P_{light}^{\alpha}$, in which $\alpha$ is the exponential factor,

and when $\alpha = 1$ indicates that there is no bimolecular recombination in the devices where all free charge carriers can be completely swept out and then collected by electrodes. As shown in Supplementary Fig. 17, the $\alpha$ values for AQx-2-, AQx-6- and AQx-8-based devices are 0.940, 0.955 and 0.938, respectively. The superior exciton dissociation and charge extraction probability and reduced bimolecular recombination in AQx-6-based OSC contributes to high $J_{SC}$ and FF.

**Energy loss analysis**
To further in-depth understand the variation of $V_{OC}$, the detailed energy loss analysis in three AQx-based OSCs was performed. The $E_{loss}$ can be quantified via SQ theory and is divided into three parts:

$\Delta E_{loss} = \Delta E_1 + \Delta E_r + \Delta E_{nr} = (E_g - qV_{OC,SQ}) + (qV_{OC,SQ} - qV_{OC,rad}) + (qV_{OC,rad} - qV_{OC})$[72,73], where $\Delta E_1$ is inevitable radiative loss above bandgap, $\Delta E_r$ and $\Delta E_{nr}$ are radiative and nonradiative loss below bandgap, $V_{OC,SQ}$ and $V_{OC,rad}$ is the $V_{OC}$ from SQ limit and radiative limit. In generally, $\Delta E_1$ is determined by the intrinsic property of materials and relates to their bandgap, the larger bandgap will produce higher $\Delta E_1$. The $\Delta E_r$ relates to absorption edge broadening and it is negligible in state-of-the-art OSCs. The $\Delta E_{nr}$ is derived from the difference between the nonideal EQE in realistic OSCs and the ideal EQE from SQ theory ($EQE_{SQ} = 1$ for $E > E_g$ and 0 for $E < E_g$), indicating the existing of nonradiative decay process. The phonon-mediated nonradiative relaxation from the local excited (LE) or charge-transfer (CT) states to the ground states is affected by the electron–phonon coupling strength that relates to molecular motion, which needs to be suppressed and is the key factor to reach a low $\Delta E_{loss}$.

The $E_g$ was calculated from the intersection point of relevant emission and external quantum efficiency spectra, as shown in Supplementary Fig. 18 and these for the D18:AQx-2, D18:AQx-6 and D18:AQx-8 based devices are 1.397 eV, 1.410 and 1.406 eV, corresponding to the $E_{loss}$s are 0.529, 0.518 and 0.493 eV, respectively. The detailed energy loss for the three devices is summarized in Supplementary Table 8. The AQx-2-, AQx-6 and AQx-8-based devices deliver almost identical $\Delta E_1$ values of 0.265, 0.266 and 0.266 eV, respectively, deriving from the almost identical bandgap among three blend films. Generally, the $\Delta E_{nr}$ was estimated from electroluminescent (EL) measurement (Supplementary Fig. 19), which equals to $-(kT/q)\ln(EQE_{EL})$ in value, where $k$ is the Boltzmann constant, $T$ is Kelvin temperature and $EQE_{EL}$ is the external quantum efficiency of electroluminescence[47]. As exhibited in Fig. 3d, AQx-2-, AQx-6- and AQx-8-based devices exhibit $EQE_{EL}$ of $3.29 \times 10^{-4}$, $4.05 \times 10^{-4}$ and $8.65 \times 10^{-4}$, respectively, corresponding to $\Delta E_{nr}$ values of 0.206, 0.198 and 0.178 eV, which is excellently in agreement with the assumption based on theoretical calculation. Considering the $\Delta E_{nr}$ may affect by the formation of triplet exciton ($T_1$), transient absorption (TA) in the near-infrared region was carried out to assess the formation of $T_1$ state in AQx-based systems. In the AQx-based blend films, an excited-state absorption (ESA) at ~1450 nm emerged after 100 ps (Supplementary Fig. 21), which can be ascribed to the $T_1$ state of the AQx-type acceptor. The ESA of $T_1$ state emerges less pronouncedly in three AQx-based blend films, suggesting the $\Delta E_{nr}$ in AQx systems is not primarily caused by the formation of $T_1$ state but rather controlled by the effect of electron-photon coupling. From AQx-2 to AQx-6 and AQx-8, the increased alkyl side chain length produces decreased FVR and restricted molecular motion with slow $k_{nr}$, which indicates a reduction in electron-photon coupling strength and then allows reduced $\Delta E_{nr}$ in OSCs. The diminution of $\Delta E_{nr}$ greatly promotes the improvement of $V_{OC}$, thus causing gradually increased $V_{OC}$ from AQx-2 to AQx-6, and then to AQx-8. The $\Delta E_r$ that comes from the equation of $\Delta E_r = E_{loss} - \Delta E_1 - \Delta E_{nr}$ for AQx-2-, AQx-6- and AQx-8-based cells are 0.059, 0.055 and 0.049 eV, indicating nearly identical radiative recombination loss below bandgap for three devices. Urbach energy of three devices was calculated from highly sensitive EQE (sEQE) throughout exponential fitting, as show in Supplementary Fig. 19, the small Urbach energy of 22-24 meV in AQx-based OSCs suggests low energetic disorder and that has been proved to be of great importance for high performance[74]. Moreover, the CT energy levels were estimated by fitting reduced EL and EQE curves (Supplementary Fig. 18)[75], those for AQx-2-, AQx-6- and AQx-8-based devices are 1.364, 1.379 and 1.372 eV, affording similar $\Delta E_{LE-CT}$s of 0.033, 0.031 and 0.034 eV, respectively. The small $\Delta E_{LE-CT}$ values in three devices mean an equally strong LE-CT hybridization and favor to improved emission efficiency, causing suppressed nonradiative decay in CT states via an intensity-borrowing mechanism[76]. The above results imply that prolonged alkyl side chain will allow low $E_{loss}$ and thus higher $V_{OC}$ in OSCs as what we speculated from the theoretical calculation, that is the proliferation of alkyl chain length can reduce FVR and restrict molecular motion via suppressed electron–phonon

coupling strength, which reveals in-depth mechanism of nonradiative decay process.

## Morphology characterization

In addition, morphology is a crucial factor for reaching high-performance OSCs and the relevant measurements contribute to understand performance difference in varied devices[77]. The alteration of alkyl side chain will affect the miscibility between D18 and AQx-type acceptors, making difference in the formation of phase separation. The contact angle measurements of neat donor and acceptor films were performed, the corresponding results are shown in Supplementary Fig. 22 and the relevant data are summarized in Supplementary Table 9. With the increase in alkyl side-chain length, AQx-6 and AQx-8 exhibit reduced surface energies (γ) of 40.35 and 40.61 mN m$^{-1}$ compared to AQx-2 (42.00 mN m$^{-1}$). The miscibility between donor and acceptor can be evaluated by the Flory–Huggins interaction parameter (χ), in which χ is defined as $\chi = K(\sqrt{\gamma_D} - \sqrt{\gamma_A})^2$. The χ values are $0.11\,K$ for D18:AQx-2, $0.04\,K$ for D18:AQx-6 and $0.05\,K$ for D18:AQx-8. The lower χ vaules in D18:AQx-6 and D18:AQx-8 imply favorable miscibility between D18 and AQx-6/AQx-8 with prolonged alkyl chain length, which is beneficial for the formation of desired phase separation. The surface morphologies of three blend film were investigated by atomic force microscopy (AFM), as shown in Supplementary Fig. 23. The D18:AQx-2 and D18:AQx-6 blend films presented a root-mean-square surface roughness ($R_q$) of 1.33 and 1.22 nm, respectively, which is probably related to the strong crystallinity of acceptor materials. Whereas the film based on D18:AQx-8 with weaker π–π packing gives a more smooth, uniform surface and a relatively low $R_q$ values of 1.02 nm. In addition, the fibers in D18:AQx-6 based blend film surface help to magnify donor/acceptor interface areas, leading to more efficient exciton dissociation as well as charge transport. Moreover, the resonant soft X-ray scattering (RSoXS), transmission electron microscopy (TEM) and grazing-incidence wide angle X-ray scattering (GIWAXS) measurements were also performed to analyze bulk morphology[78]. RSoXS is used to investigate the phase separation information in AQx-based blend films. As shown in Supplementary Fig. 24, D18:AQx-2 gives the largest domain size of 30.72 nm, while D18:AQx-6 and D18:AQx-8 exhibit phase separation with domain sizes of 21.55 and 22.10 nm, respectively, which is close to the optimal domain size of 10–20 nm generally accepted for OSCs. Such a result can be further confirmed by the TEM measurements (Supplementary Fig. 23). The GIWAXS results are shown in Fig. 4a, b and the detailed data were summarized in Supplementary Tables 10-11. In the neat film, all three acceptors show preferred face-on orientation, verified by the strong π–π packing (010) peaks in the out-of-plane (OOP) direction and lamellar (100) peaks in the in-plane (IP) direction. Nevertheless, the AQx-8 with the longest alkyl chain exhibits weak (010) π–π packing peaks in the IP direction, suggesting that there is partial edge-on orientation in AQx-8 thin film, which proclaims diverse aggregation behavior in AQx-8 from the other acceptors. From AQx-2 to AQx-8 with the lengthened alkyl chain, the π–π stacking peak (010) gradually shifts to the lower q region, with the specific location at 1.742, 1.737 and 1.704 Å$^{-1}$, corresponding to the π–π stacking distances of 3.605, 3.615 and 3.685 Å, respectively, and demonstrating that the crystallinity is disturbed by the long alkyl chain, which has also been observed in previous reports[22]. In Supplementary Fig. 25, D18 exhibits a clear (010) peak in the OOP direction, suggesting the favorable face-on orientation. Moreover, the (001) diffraction peak at ~0.5 Å in the IP direction can be used to access crystallization feature of D18[35]. Such a peak was observed in the D18:AQx-2 and D18:AQx-6 blend films but disappeared in D18:AQx-8 (Fig. 4b) suggests that D18 can retain a good crystallinity in AQx-2- and AQx-8-based blends, contributing to improved charge transport. For all blend films, the favorable face-on orientation is maintained, which allows the free charge carriers can be efficiently transported to electrodes in the vertical direction. In the OOP

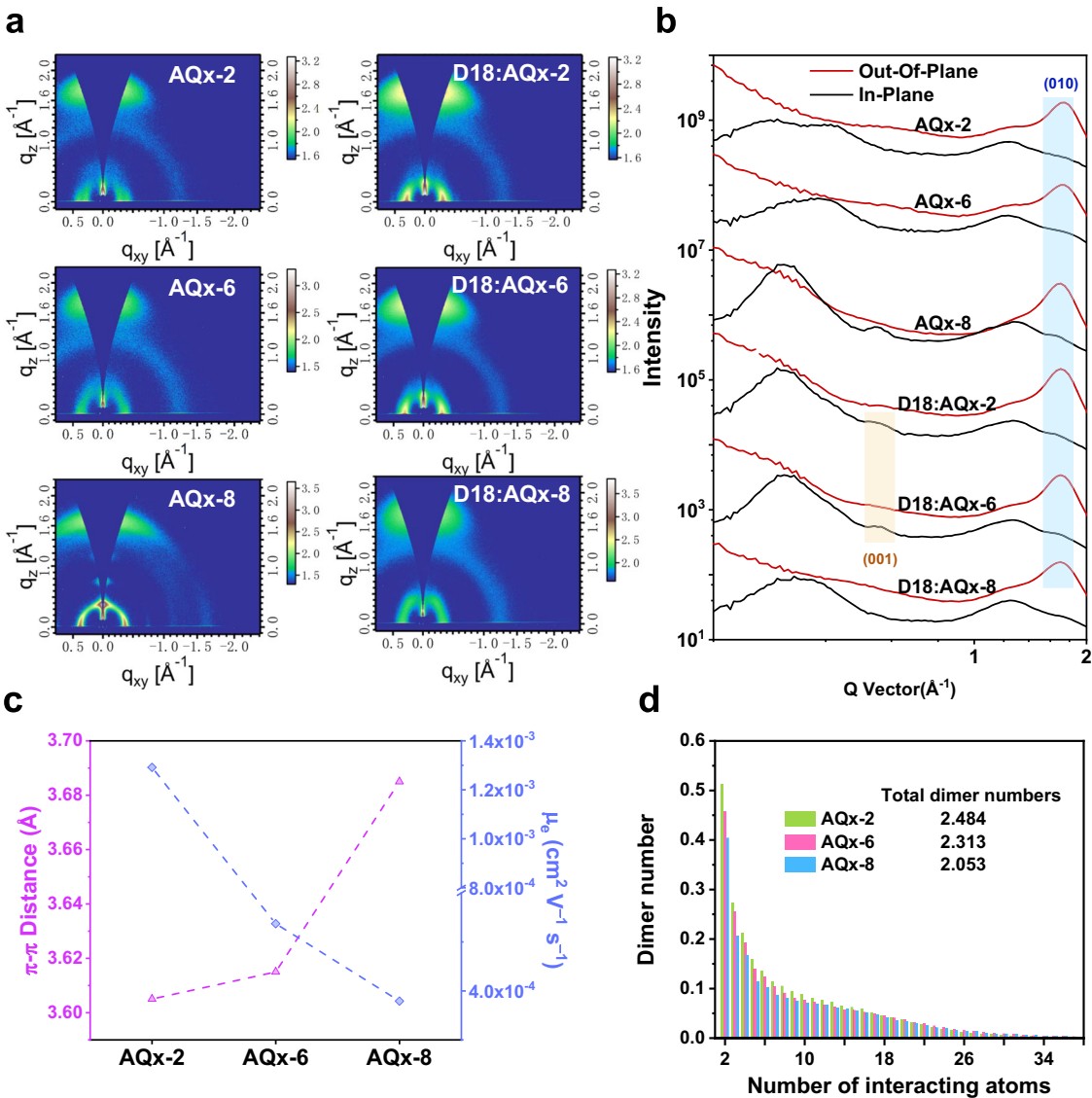

**Fig. 4 | Morphology investigations. a** GIWAXS 2D patterns for AQx-2, AQx-6 and AQx-8 pristine films and the corresponding blend films. **b** The corresponding line-cuts of GIWAXS patterns. **c** $\pi-\pi$ distance and electron mobility. **d** Dimer numbers (per molecule) for AQx-2, AQx-6 and AQx-8 pristine films.

direction, the $\pi-\pi$ packing distances for AQx-2-, AQx-6- and AQx-8-based blend films are 3.662, 3.666 and 3.664 Å, with relevant crystal coherence lengths (CCLs) are 25.01, 20.11 and 17.66 Å, respectively. The longer CCLs in AQx-2- and AQx-6-based films are conducive to improving charge transport and suppressing charge recombination, while the reduced CCL in AQx-8 may induce more charge recombination. The charge transport property of the three devices can be investigated by the space-charge-limited-current (SCLC) measurement. As shown in Fig. 4c, the electron mobilities ($\mu_e s$) in pristine AQx-2, AQx-6 and AQx-8 are $1.29 \times 10^{-3}$, $5.37 \times 10^{-4}$ and $3.79 \times 10^{-4}$ cm$^2$ V$^{-1}$ s$^{-1}$, the higher $\mu_e$ is consistent with the stronger crystallinity of AQx-2 and AQx-6. For the AQx-2-, AQx-6- and AQx-8-based devices, the $\mu_e s$ and hole mobilities ($\mu_h s$) are $4.95 \times 10^{-4}$, $4.75 \times 10^{-4}$, $3.66 \times 10^{-4}$ cm$^2$ V$^{-1}$ s$^{-1}$ and $2.54 \times 10^{-4}$, $3.16 \times 10^{-4}$, $1.86 \times 10^{-4}$ cm$^2$ V$^{-1}$ s$^{-1}$, respectively, and that corresponds to $\mu_e/\mu_h$ ratios of 1.95, 1.50 and 1.97. The higher charge mobilities in AQx-2- and AQx-6-based blends are in favor of superior charge transport, in comparison to those in AQx-8-based blend, which is consistent with the results of GIWAXS. Besides, the low $\mu_e/\mu_h$ ratio in AQx-6-based blend suggests more balance charge transport, leading to less recombination and the highest FF.

To deeply investigate the influence of alkyl side chain, the calculations on the dimer number for individual rigid backbones were performed, in which dimer number relates to the contact between interacting backbones. The total dimer numbers over 2 indicates good intermolecular connectivity, which is a necessary precondition for efficient charge transport. As presented in Fig. 4d, the total dimer numbers for AQx-2, AQx-6 and AQx-8 film are 2.484, 2.313 and 2.053, respectively, implying the existing of continuous charge transport channels. In addition, the decreasing trend of dimer number with increased alkyl side-chain length in AQx-type acceptors indicates prolonged alkyl side chain will undermine the continuous aggregation of backbones. The $\pi-\pi$ packing can be accessed by the theoretical calculations on the distribution of average dimer number (over 8 backbones interacting atoms) for per molecule. As shown in Supplementary Fig. 26, AQx-2, AQx-6 and AQx-8 show the total interactions for $\pi-\pi$ stacking of 1.074, 1.036 and 0.973, indicating the high crystallinity exists in AQx-2 and AQx-6, which is in accord with experimental results. The AQx-2 with shortest alkyl chain shows large separation with less D/A interface for exciton dissociation, while the extensively extended alkyl side chain in AQx-8 generates distinct weakened $\pi-\pi$ packing, leading to more charge recombination. On the

contrary, AQx-6 with subtle prolonged alkyl side chain can form suitable domain size for efficient exciton dissociation and maintain compact $\pi-\pi$ packing for the high rate of charge transport. Therefore, there is a delicate balance between low FVR (for reduced nonradiative decay rate) and ideal morphology (desired phase separation for exciton dissociation and strong crystallinity for efficient charge transport), inspiringly, precisely alkyl side-chain modulation can allow a subtle balance and toward higher performance.

## Charge dynamics

Furthermore, the charge dynamics including exciton transport and charge generation have been in-deep investigated for better understanding of three devices. The TA measurement was carried out to analyze the charge-transfer dynamics in the *D/A* interface. The 800 nm pump beam was selected to excite NFA in three blends, which can prevent donor from being excited. Figure 5 and Supplementary Figs. 27–28 show the TA results for neat and blend films. The pristine acceptor films exhibit the ground-state bleaching (GSB) peaks at ~730 and ~840 nm and the excited-state absorption (ESA) signal of local excited (LE) state at ~920 nm, which can be observed in blend films. After 10 ps, the ESA signals emerging at ~770 nm, which can be assigned to the charge separation (CS) state in the blend films, indicting the onset of the exciton dissociation process[79]. The two bleach peaks at ~560 and ~590 nm can be ascribed to the GSB of donor, considering there are on singles in neat acceptor films. Herein, the GSB peak at 590 nm was chosen to analyze the hole transfer kinetics, as shown in Fig. 5d. All three blends give quick emergence of the bleach signal of D18 (Supplementary Fig. 28), indicating an ultrafast hole transfer process from acceptor to donor in three blends. Such phenomenon was also observed in other highly performed OSCs. To get a better understanding, the hole transfer can be evaluated by fitting the GSB peak of donor at 590 nm with a double exponential function, thus came: D18:AQx-2 ($\tau_1 = 2.94 \pm 0.12$ ps, $\tau_2 = 49.23 \pm 0.83$ ps), D18:AQx-6 ($\tau_1 = 2.24 \pm 0.08$ ps, $\tau_2 = 27.40 \pm 0.42$ ps) and D18:AQx-8 ($\tau_1 = 2.24 \pm 0.10$ ps, $\tau_2 = 31.71 \pm 0.45$ ps), respectively, where $\tau_1$ is related to exciton dissociation rate at the D/A interface and $\tau_2$ represents the time for photo-induced exciton diffuse to D/A

interface[80]. The slower exciton diffusion rate in D18:AQx-2 may be related to its large domain size, while D18:AQx-6 and D18:AQx-8 possess refined phase separation, which can contribute the exciton diffusion from intradomain into D/A interface at higher rate. The photoluminescence (PL) quenching measurements were carried out to study the charge-transfer efficiency (Supplementary Fig. 29). All of three blend films exhibit highly efficient electron transfer efficiency due to the sufficient driving force from LUMO energy offset. When NFA was selected excited, the AQx-2-, AQx-6-, and AQx-8-based bends display the hole transfer efficiency of 94.3, 94.4 and 91.0%, respectively. These results show that D18:AQx-6-based OSC delivers highest exciton dissociation and charge-transfer efficiency than those in AQx-2- and AQx-8-based OSCs, contributing to the formation of superior $J_{SC}$ and FF[81,82].

## Discussion

In this work, we demonstrated that efficient OSCs with reduced $\Delta E_{nr}$ can be enabled by suppressed electron–phonon coupling, and proposed that the FVR can act as the indicator to evaluate molecular motion in solid state associated with nonradiative decay process. We developed a series of AQx-type acceptors, AQx-2, AQx-6 and AQx-8, with gradually extended alkyl side-chain length based on rational molecular design. Theoretical and experimental results confirmed that prolonged alkyl side chain contributes to decreased FVR in aggregation states. The restricted molecular motion in limited free volume provides suppressed nonradiative decay rate and then offers reduced $\Delta E_{nr}$ in OSCs. Energy loss analysis suggests that AQx-6 and AQx-8 possess lower $\Delta E_{nr}$ and elevated $V_{OC}$ values, compared to that in AQx-2 with the shortest alkyl side chain. Whereas the variation of alkyl side-chain length also affects the nanomorphology in blend films. For one thing, the increased alkyl side-chain length in AQx-type acceptors induces improved miscibility between NFA and donor, leading the formation of small phase separation that provides efficient exciton dissociation. For another, the long alkyl chain will affect the ordering $\pi-\pi$ stacking in film, causing decreased mobilities and increased charge recombination. Inspiringly, the asymmetrical alkyl side chain in AQx-6 endows it with good miscibility with D18 to from suitable

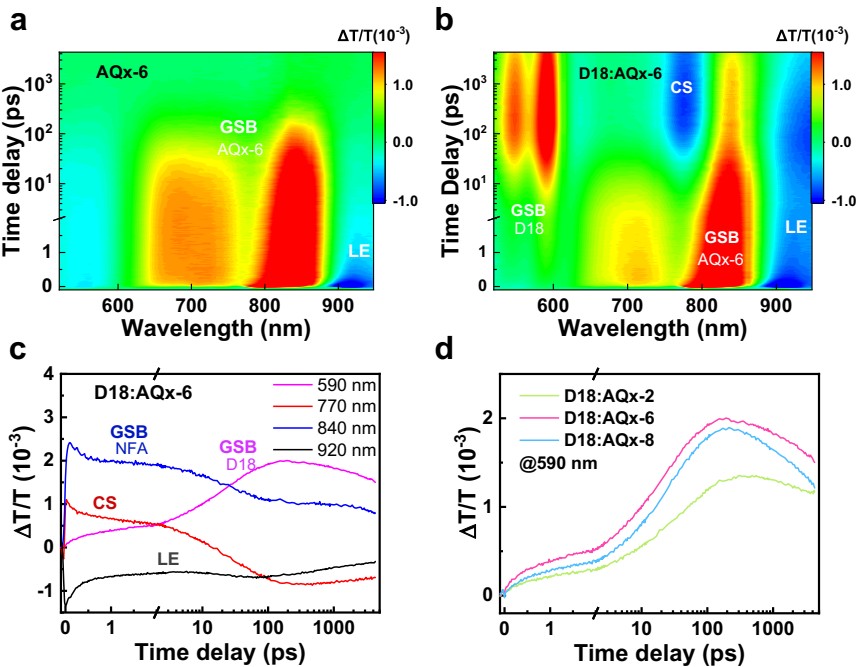

**Fig. 5 | Transient absorption.** TA spectra of AQx-6 neat film (**a**) and D18:AQx-6 blend film (**b**). **c** TA traces of GSB signal for D18: 590 nm, the CS signal for D18:AQx-6 blend: 770 nm, the GSB (840 nm) and LE signals (920 nm) of AQx-type acceptor.

**d** Kinetic traces of D18:AQx-2, D18:AQx-6 and D18:AQx-8 at indicated wavelength (590 nm).

separation and compact $\pi$–$\pi$ packing, which is of great importance for efficient charge generation and less charge recombination. As a result, the reduced $\Delta E_{nr}$ and favorable nanomorphology for efficient exciton dissociation and charge transport are simultaneously obtained in AQx-6 via precisely unilateral proliferation of alkyl side chain. Ultimately, the OSCs based on AQx-6 achieved a champion PCE of 18.6% (certified PCE: 18.4%), which is one of the highest PCEs among bulk-heterojunction binary OSCs. Tuning hybridization of LE and CT states via modulating energy level, reducing reorganization energy or adjusting electrostatic potential distribution that minishes $\Delta E_{nr}$ has been proved to be effective strategies, which is ascribed to the restrained nonradiative decay process[41,48,58,83–87]. This work generates a new direction that decreases electron–phonon coupling associated with molecular motion to slow down $k_{nr}$, contributing to enlightening research to construct low $\Delta E_{nr}$ via a rational design strategy, which is of great significance to the further development of OSCs.

## Methods

### Materials
D18 was purchased was purchased from Derthon Optoelectronic Materials Science Technology (Shenzhen, China). AQx-2 was synthesized according to reported literature[63]. The synthesis route of AQx-6 is shown in Scheme S1, AQx-8 was synthesized according to AQx-2.

### Theoretical calculation
Molecular dynamics simulations were conducted with GROMACS-4.6.7 software package. All DFT calculations were experimented with the Gaussian 16 codes. Setting of Force Field Parameters: for both the solute AQx-2, AQx-6, AQx-8 and the solvent chloroform force fields, their atom types, intramolecular and intermolecular interaction parameters were provided in the general AMBER force field (GAFF). Their atomic charges were calibrated using restrained electrostatic potential (RESP) obtained by density functional theory (DFT) calculations in vacuum at ωb97xd/cc-pvtz level. The solute's torsional potentials between their central D and A units were fitted with DFT calculated potential energy, which used B3LYP-D3(BJ) functional with dispersion correction of DFT-D3 and a BJ damping under 6−31G** basis set. MD Simulation Protocol: The protocol consists of an NPT ensemble equilibrium molecular dynamics simulation for 50 ns of a solvated AQx and a quasi-equilibrium molecular dynamics approach to model the solvent evaporation and thin-film formation process of AQx. Both of the simulations proceeded under 300 K, 1 bar to align with experiment environments. The quasi-equilibrated system was a mixture of 60,000 chloroform and 200 conformations of AQx which was extracted during the NPT equilibrated simulation. The system was first equilibrated for 10 ns and then evaporated 100 random solvents for each 100 ps. When the thin film formed, it was annealed to 500 K and quenched back to 300 K. This annealing process is repeated until the system reached equilibrium. The protocol was followed for AQx-2, AQx-6 and AQx-8.

### Single-crystal growth
Single crystal of AQx-2 was grown by the liquid diffusion method at room temperature. An appropriate amount of diethyl ether is transferred to a concentrated chloroform solution, which will form crystals over time. Single-crystal diffraction data was collected by single-crystal X-ray diffractometer (XtaLAB PRO 007HF(Mo)). The X-ray crystallographic coordinates for structures reported of AQx-2 has been deposited at the Cambridge Crystallographic Data Centre (CCDC), under deposition numbers 2161927.

### Device fabrication
The device with a conventional structure: ITO/PEDOT:PSS/D:A/PDINN/Ag. The substrates were sequentially cleaned by detergent, water, acetone, ethyl alcohol for 30 min each step. The PEDOT:PSS (30 nm) was spin-coated on pre-cleaned ITO substrates after ITO substrates were treated with 30 min oxygen plasma. Then the substrates were heated at 150 °C for 20 min under air condition. The substrates were then transferred into a nitrogen-filled glove box. The weight ratio of D18:acceptor is 1:1.2 and apply chloroform as solvent. The D18:acceptor blend solution with a total concentration of 9.5 mg/mL was heated at 100 °C till completely dissolved. Then the blend solution was spin-coated on PEDOT:PSS by 2000 rpm, the thickness of the active layer is 110 nm. And the D18:accepter was heated at 80 °C for 10 min. The PDINN in ethyl alcohol (1 mg/ml) was spin-coated on the active layer by 3000 rpm. The Ag (100 nm) was deposed onto the substrates by vacuum evaporation. Shadow masks were used to define the OSC active area (0.04 cm²) of the devices.

### Device characterization
The current density–voltage ($J$–$V$) characteristics of unencapsulated photovoltaic devices were measured under $N_2$ using a Keithley 2400 source meter. The illumination of AM 1.5 G (100 mW cm⁻²) was achieved by a XES-70S1 solar simulator (SAN-EI Electric Co., Ltd, AAA grade, 70 mm × 70 mm photo-beam size). The external quantum efficiency (EQE) was performed using certified incident photon to current conversion efficiency equipment from Enlitech (Taiwan).

### SCLC mobility measurement
Space-charge-limited currents were tested in hole-only devices with a structure of ITO/PEDOT:PSS/active layer/Au and electron-only devices with a configuration of ITO/ZnO/active layer/PDINN/Ag. The devices were prepared following the same procedure described in the experimental section for photovoltaic devices, except for the metal electrode. The hole and electron mobilities were calculated as follows:

$$J = \frac{9\varepsilon_0\varepsilon_r\mu_0 V^2}{8L^3} \quad (1)$$

Where $J$ is the current density, $\mu_0$ is the zero-field mobility, $\varepsilon_0$ is the permittivity of free space, $\varepsilon_r$ is the relative permittivity of the material, $V$ is the effective voltage, and L is the thickness of the active layer.

### Morphology characterization
Atomic force microscopy (AFM) images of the thin films were obtained on a NanoscopeIIIa AFM (Digital Instruments) operating platform in tapping mode. The samples were prepared by spinning coated the active layers on the PEDOT:PSS layer.

Transmission electron microscopy (TEM) observation was performed on JEOL 2200FS at 80 kV accelerating voltage. The samples for electron microscopy were prepared by dissolving the PEDOT:PSS layer using water and transferring the floating active layer to the TEM grids. Grazing-incidence wide angle X-ray scattering (GIWAXS)was performed at the National Center for Nanoscience and Technology, China (NCNST). Thin film samples were spin-casted on to PEDOT:PSS covered SiO₂ wafers.

### Energy loss analysis
Highly sensitive EQE was measured using an integrated system (PECT-600, Enlitech), where the photocurrent was amplified and modulated by a lock-in instrument. $EQE_{EL}$ measurements were performed by applying external voltage/current sources through the devices (ELCT-3010, Enlitech). All the devices were prepared for $EQE_{EL}$ measurements according to the optimal device fabrication conditions.

### Spectroscopic measurements
Transient Absorption (TA) Spectroscopy measurements were performed on an ultrafast TA spectrometer (Harpia-TA, Light Conversion). An 1030 nm pulse with a repetition rate of 54 kHz was generated by a Yb:KGW laser. Then the pulse was separated into two parts by a beam splitter. One part was coupled into an optical parametric amplifier to

generate the pump pulses at various wavelength. The other part was focused onto a sapphire plate to generate supercontinuum white light as the probe beams. The time delay between pump and probe was controlled by a motorized optical delay line with a maximum delay time of 3.6 ns. The pump pulse is chopped by a mechanical chopper and then focused on to the mounted sample with probe beams. The probe beam was collimated and focused into a spectrometer with CCD sensor. The photoluminescence (PL) and photoluminescence quantum yield (PLQY) measurements were performed in Analytical Instrumentation Center of Peking University. The PL spectra were recorded with lifetime and steady state spectrometer (FLS980). The PLQY were detected with lifetime and steady state spectrometer (FLS980), which was coupled with OXFORD microstat. The time-resolution photoluminescence (TRPL) measurement was performed by Femtosecond Ultrafast Spectroscopy (Astrella-Opera solo).

## Reporting summary

Further information on research design is available in the Nature Portfolio Reporting Summary linked to this article.

## Data availability

The authors declare that the data supporting the findings of this study are available within the paper and its supplementary information files. The single-crystal data in this study have been deposited at the Cambridge Crystallographic Data Centre (CCDC), under deposition numbers 2161927. The single-crystal data can be obtained free of charge from the Cambridge Crystallographic Data Centre via www.ccdc.cam. ac.uk/data_request/cif. Source data are provided with this paper.

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

## Acknowledgements

This work was supported by the National Key R&D Program of China (2019YFA0705900 (X.Z.) and 2017YFA0204701 (X.Z.)), the National Natural Science Foundation of China (52225305 (X.Z.), 22175187 (X.Z.), 22171273 (S.X.), U2032112 (W.Z.), 21905163 (F.L.) and 22173108 (Y.Y.)), the Open Funding Project of State Key Laboratory of Organic-Inorganic Composites (oic-202201002 (X.Z.)), the International Partnership Program of Chinese Academy of Sciences (027GJHZ2022036GC (X.Z.)) and the Youth Innovation Promotion Association CAS (No. 2020031 (S.X.)). The authors thank Yanni Ouyang and Prof. Chunfeng Zhang for their help with transient absorption measurements, and Jinqiu Xu, Jingming Xin and Prof. Wei Ma for their help with Resonant soft X-ray scattering measurements.

## Author contributions

X.Z. conceived and supervised the project. Y.J., Y.L. and F.L. contributed equally to this work. S.X. synthesized the AQx-2, AQx-6 and AQx-8; S.L. synthesized the AQx-6; Y.J. and F.L. fabricated, optimized the devices and performed the device characterizations, which were supervised by X.Z.; Y.L. carried out theoretical calculation, which were supervised by Y.Y.; W.S. performed transient absorption, which were supervised by W.Z.; W.W. performed the EL and FTPS-EQE experiments, which were supervised by J.H; W.L draw the figures. Y.J., Y.L., F.L., S.X., Y.Y. and X.Z. discussed the results and substantially contributed to the preparation of the manuscript.

## Competing interests

The authors declare no competing interests.
