## [Peer Review File · Nature Communications]

REVIEWER COMMENTS

Reviewer #1 (Remarks to the Author):

Nonradiative energy loss plays an essential role in determining the OSCs performance, which is affected by the strong electron-phonon coupling in organic materials. In this manuscript, the authors investigated the relationship between molecular aggregation behavior and electron-phonon coupling via AQx-type acceptors with different alkyl side-chain length, which provides a new insightful view on how to reduce nonradiative energy loss in terms of aggregation structure regulation. The results confirm that low free volume ratio in aggregated state can effectively suppress nonradiative decay process, leading to a significant reduction in nonradiative energy loss. Moreover, a very high PCE of 18.6% was achieved in D18:AQx-6-based binary OSCs. Overall, the manuscript is well written and organized, and can be highly recommended for the publication in Nature communications after the points noted below have been addressed in a revision.

1. Compared to AQx-2 and AQx-6, AQx-8 pristine film shows faster radiative rate and the author ascribed it to the reduced H-aggregation in AQx-8 film, but the authors did not give an explanation why AQx-8 delivers reduced H-aggregation that differs from AQx-2 and AQx-6, whether it is associated with the longest alkyl side-chain in AQx-8?

2. In the single-crystal analysis part, the authors assigned the aggregation between end groups to J-aggregation and the aggregation between backbones to H-aggregation, which is less convincing, could the authors give more results to support it?

3. The author adopted alkyl side-chain engineer to suppress nonradiative energy loss with the precondition of high charge generation efficiency and eventually achieved an inspiring PCE of 18.6% in binary AQx-6-based device. Considering the PCE of 18.6% is the highest reported PCE among binary OSCs to date, could the authors provide the certified PCE to confirm the high performance of D18:AQx-6-based device?

4. TA measurement is an important and common approach to get a better understanding of charge dynamic, which also performed in this manuscript to investigated different exciton dissociation behavior among AQx-2, AQx-6 and AQx-8. However, there are some problems exist in the analysis of TA spectrum as below:

(1) The manuscript claims that "Three blends all give a quick emergence of bleach signal, corresponding to the fast decayed GSB peak of acceptors at 840 nm, indicating ultrafast hole transfer process from acceptor to donor in three blends." However, the acceptor's GSB signal always appears immediately

when excited acceptor, regardless of whether it is in a blend or a pure film. The author should clarify this point or provide more precise language.

(2) The manuscript claims that “the negative peaks at ~900 nm can be assigned to the excited-state absorption (ESA) of acceptors.” However, the ~900nm peak should be positive. Additionally, to confidently assign this signal to the acceptor's ESA, the TA spectrum of pure films is necessary and should be included in the manuscript.

(3) The manuscript shows a positive signal at ~780nm that appears after 10 ps in all blends. The author should assign this signal in the manuscript.

5. Since the AQx-6 and AQx-8 were firstly reported in this manuscript, both of them should be confirmed by ¹H NMR, ¹³C NMR and HR-MS, thus, please provide the ¹³C NMR of AQx-8.

6. Some mistakes should be corrected and please carefully check the manuscript to prevent other errors.

Page 3, line 98, “AQx-6 and AQx-8 allows” should be “AQx-6 and AQx-8 allow”.

Page 3, line 104, “The τ value” should be “The τ values”.

Page 3, line 110, “agrees well with” should be “agree well with”.

Reviewer #2 (Remarks to the Author):

In this manuscript, the authors designed and synthesized a series of AQx-type acceptors with varied alkyl side-chain length, and proposed that alkyl side-chain can effectively modulate aggregation in solid state, namely free volume ratio, and then affect the nonradiative decay process. Both the theoretic and experimental results demonstrated that long alkyl side chain can provide low free volume ratio and suppress molecular motion. By balancing the nonradiative energy loss and charge transport property, OSCs based on AQx-6 achieved the highest PCE of 18.6%, which is the highest performance among reported binary OSCs. This finding provides a unique perspective on the suppression of nonradiative energy loss in organic photovoltaic technique, which should be of high interest for OSC community. Therefore, I strongly recommend this manuscript to be accepted for publication in Nature communications after addressing the following issues.

1. Compared to inorganic solar cells, larger nonradiative energy loss limits the efficiency promotion of OSCs. Many efforts have been devoted to explore how to reduce nonradiative energy loss and then

achieve higher performance. In contrast to early studies, the authors have got a very impressive new finding, and I suggest them to more clearly express the difference and significance of this finding.

2. In generally, charge transport property is closely related to the aggregation behavior in solid state, and 3D packing is in favors to form 3D charge transport channels (Nat. Energy 2021, 6, 605). As the authors described in the manuscript, AQx-type acceptor possess 2D packing in single-crystal, and previous study demonstrated that Y6 shows 3D packing. However, D18:AQx-6 delivers higher performance than that of D18:Y6. Can the authors give more explanations on it?

3. By precisely extending alkyl chain length, AQx-6 can simultaneously exhibit suppressed electron-phonon coupling for low energy loss, high charge transport property and efficient charge generation efficiency. If possible, the performance of AQx-6-based cells should be verified by a trusted third authority.

4. In the part of morphology characterization, the experimental results suggested that there is a great difference among three devices. As the authors mentioned in the manuscript, D18:AQx-2-, D18:AQx-6- and D18:AQx-8-based blends show different phase separation behaviors. Considering the alkyl side-chain will affect the material solubility, the miscibility between donor and acceptors should be studied.

5. Since the AQx-6 and AQx-8 were firstly reported in this work, the full name of AQx-6 and AQx-8 should be provided.

6. The thermogravimetric analysis (TGA) measurements of three acceptors should be performed to study the material thermostability.

7. While the device optimization process of D18:AQx-6 was summarized in Supplementary information, the author did not give the optimization process of D18:AQx-2 and D18:AQx-8. The related data also need to be provided.

8. Due to the weaker π - π stacking in AQx-8 than AQx-2 and AQx-6, AQx-8 presented inferior electron mobility, causing more severe bimolecular recombination, which was verified by the JSC-Plight relationship. The recombination mechanism should be discussed in the main text.

Reviewer #3 (Remarks to the Author):

I am leaning towards rejection, and suggest that the revised manuscript be submitted to a more specialized journal. The 18% PCE is impressive but it is not a record. Side chain engineering to improve efficiency is not new. Improving efficiency using asymmetric acceptors to tune balance between charge transfer and recombination has already been done before.

I struggled a bit to get the central message of this manuscript by Yuanyuan Jiang et al. The arguments need to be better articulated. They are connecting a lower free volume ratio (FVR) with lower $\Delta(\text{Enr})$ and a higher Voc. The connection seems to be non-linear and they have not discussed other competing

processes such as triplet exciton formation for non-radiative energy losses. The authors later discuss about connections between crystallinity to explain mobilities, J_{sc} and FF. However, they have not done any texture analysis nor have they extracted numbers for the relative degree of crystallinity from the GIWAXS data. As a sidenote, the study would be more complete if they also showed GIWAXS data from neat D18. The authors haven't also made any connection between FVR and the crystallinity. Having RSoXS would further help to characterize the morphology and could further explain the variations in device performance.

Detailed descriptions of our replies are given below, where the reviewers' original comments appear in **blue**; our replies, in **black**; and the modifications to the manuscript, in **red**.

Response to Reviewer 1:

> Nonradiative energy loss plays an essential role in determining the OSCs performance, which is affected by the strong electron-phonon coupling in organic materials. In this manuscript, the authors investigated the relationship between molecular aggregation behavior and electron-phonon coupling via AQx-type acceptors with different alkyl side-chain length, which provides a new insightful view on how to reduce nonradiative energy loss in terms of aggregation structure regulation. The results confirm that low free volume ratio in aggregated state can effectively suppress nonradiative decay process, leading to a significant reduction in nonradiative energy loss. Moreover, a very high PCE of 18.6% was achieved in D18:AQx-6-based binary OSCs. Overall, the manuscript is well written and organized, and can be highly recommended for the publication in Nature communications after the points noted below have been addressed in a revision.

Our response: We sincerely thank Reviewer 1 for the very nice and positive comments.

> Compared to AQx-2 and AQx-6, AQx-8 pristine film shows faster radiative rate and the author ascribed it to the reduced *H*-aggregation in AQx-8 film, but the authors did not give an explanation why AQx-8 delivers reduced *H*-aggregation that differs from AQx-2 and AQx-6, whether it is associated with the longest alkyl side-chain in AQx-8?

Our response: Thanks for the reviewer's professional comment.

We agree with the reviewer's opinion. Considering that the alkyl chain is attached on the sp^2 -hybridized nitrogen atoms in the pyrrole rings, the prolonged alkyl chain with large steric hindrance will more disturb the aggregation between backbones. The main text has been revised as follow.

“The faster radiative decay process and higher PLQY are observed in AQx-8, which can be attributed to a slightly looser π - π stacking that is caused by long alkyl side-chain, leading to a reduced *H*-aggregation.⁶⁹”

> In the single-crystal analysis part, the authors assigned the aggregation between end groups to *J*-aggregation and the aggregation between backbones to *H*-aggregation, which is less convincing, could the authors give more results to support it?

Our response: Thanks for the reviewer's good suggestion.

From the single crystal structure of AQx-2 (**Figure S8**), dimer 1 shows a large overlapped area, which trends to be regarded as the *H*-aggregation, while dimer 2 exhibit small overlapped area via the end groups, which usually ascribes to *J*-aggregation. In order to further confirm the aggregation types in AQx-2 single-crystal, we extracted the dimers from the crystal structure with optimized hydrogen positions and constrained backbone.

From the theoretical calculations (**Figure S9**), the S_1 excited state oscillation strength is zero for dimer 1, which is a typical characteristic for H -aggregation. In comparison, the oscillation strength is slight increased dimer 2, compared that in single-molecular state, which demonstrates that it is a J -aggregation. Therefore, it is can be inferred

Figure S8. The arrangement of the experimental AQx-2 single crystal.

Figure S9. The HOMO-LUMO orbital, transition dipole moments, and transition dipole moments of monomer (left), dimer 1 (middle), dimer 2 (right).

> The author adopted alkyl side-chain engineering to suppress nonradiative energy loss with the precondition of high charge generation efficiency and eventually achieved an inspiring PCE of 18.6% in binary AQx-6-based device. Considering the PCE of 18.6% is

the highest reported PCE among binary OSCs to date, could the authors provide the certified PCE to confirm the high performance of D18:AQx-6-based device?

Our response: We have provided a certified PCE of the D18:AQx-6-based device from the National Institute of Metrology (NIM), China, and the results are shown in **Figure S13**. The related data were added to the revised manuscript on page 6, as shown below:

“Besides, the certification of D18:AQx-6 performance was carried out in National Institute of Metrology (NIM), China, and a certified PCE of 18.4% with a V_{OC} of 0.874 V, a J_{SC} of 26.7 mA cm⁻² and an FF of 79.1% was obtained (**Figure S13**).”

Figure S13. The certified PCE of D18:AQx-6, obtained from National Institute of Metrology (NIM), China.

> TA measurement is an important and common approach to get a better understanding of charge dynamic, which also performed in this manuscript to investigated different exciton dissociation behavior among AQx-2, AQx-6 and AQx-8. However, there are some problems existed in the analysis of TA spectrum as below:

- (1) The manuscript claims that “Three blends all give a quick emergence of bleach signal, corresponding to the fast decayed GSB peak of acceptors at 840 nm, indicating ultrafast hole transfer process from acceptor to donor in three blends.” However, the acceptor’s GSB signal always appears immediately when excited acceptor, regardless of whether it is in a blend or a pure film. The author should clarify this point or provide more precise language.
- (2) The manuscript claims that “the negative peaks at ~900 nm can be assigned to the excited-state absorption (ESA) of acceptors.” However, the ~900nm peak should be positive. Additionally, to confidently assign this signal to the acceptor’s ESA, the TA spectrum of pure films is necessary and should be included in the manuscript.

(3) The manuscript shows a positive signal at ~780 nm that appears after 10 ps in all blends. The author should assign this signal in the manuscript.

Our response: Thank you for the valuable suggestions.

(1) We have made the relevant alteration in the revised manuscript (page 11).

“All three blends give quick emergence of the bleaching signal of D18 (**Figure S27**), indicating that an ultrafast hole transfer process from acceptor to donor in three blends. Such phenomenon was also observed in other highly performed OSCs.”

(2) The TA profiles for neat acceptor (**Figure S26**) were added in the revised manuscript and the relevant alteration was shown in page 11.

“The pristine acceptor films exhibit the ground-state bleaching (GSB) peaks at ~730 and ~840 nm and the excited-state absorption (ESA) signal of local excited (LE) state at ~920 nm, which can be observed in blend films.”

Figure S26. TA profiles (a), TA spectra at different time delay (b) and TA traces of ground-state bleaching (GSB) signals probed at 840 nm and ESA signals of local excited (LE) state probed at 920 nm (c) for AQx-2, AQx-6 and AQx-8 neat films with pump at 800 nm.

(3) The positive signal at ~780 nm can be assigned to the charge separation state, which is assured in the current investigation (*J. Am. Chem. Soc.* **2020**, *142*, 12751). The relevant discussion has been added to the manuscript (page 11).

“After 10 ps, the ESA signal emerges at ~770 nm, which can be assigned to the charge separation (CS) state in blend films, indicating the onset of the exciton dissociation process.”⁷⁹”

Figure S27. TA profiles (a), TA spectra at different time delay (b) and TA traces of GSB signal for D18: 590nm, the CS signal for D18:AQx-6 blend: 770 nm, the GSB (840 nm) and LE signals (920 nm) of AQx-type acceptor (c) for AQx-2, AQx-6 and: AQx-8 neat films with pump at 800 nm.

> Since the AQx-6 and AQx-8 were firstly reported in this manuscript, both of them should be confirmed by ¹H NMR, ¹³C NMR and HR-MS. Thus, please provide the ¹³C NMR of AQx-8.

Our response: Thank you for the professional suggestion.

The ¹³C NMR spectrum of AQx-8 was performed (**Figure S38**), which was added into Supplementary information.

“¹³C NMR (101 MHz, CDCl₃): δ 186.1, 158.9, 155.7, 155.7 154.1, 153.1, 153.1, 153.0, 152.9, 146.2, 142.8, 138.1, 136.9, 136.6, 136.6, 136.2, 135.3, 134.5, 133.2, 133.2, 130.7, 119.7, 119.5, 115.1, 115.0, 114.8, 114.6, 112.5, 112.3, 68.4, 55.6, 39.1, 31.9, 31.8, 31.6, 31.5, 31.4, 30.5, 29.9, 29.9, 29.8, 29.7, 29.7, 29.6, 29.5, 29.5, 29.5, 29.4, 29.4, 29.2, 25.6, 25.5, 22.7, 22.6, 22.5, 14.1, 14.1, 14.0.”

Figure S38. The ^{13}C NMR spectrum for AQx-8.

> Some mistakes should be corrected and please carefully check the manuscript to prevent other errors.

Page 3, line 98, “AQx-6 and AQx-8 allows” should be “AQx-6 and AQx-8 allow”.

Page 3, line 104, “The τ value” should be “The τ values”.

Page 3, line 110, “agrees well with” should be “agree well with”.

Our response: We have carefully checked the manuscript and revised the corresponding typos.

Response to Reviewer 2:

> In this manuscript, the authors designed and synthesized a series of AQx-type acceptors with varied alkyl side-chain length, and proposed that alkyl side-chain can effectively modulate aggregation in solid state, namely free volume ratio, and then affect the nonradiative decay process. Both the theoretic and experimental results demonstrated that long alkyl side chain can provide low free volume ratio and suppress molecular motion. By balancing the nonradiative energy loss and charge transport property, OSCs based on AQx-6 achieved the highest PCE of 18.6%, which is the highest performance among reported binary OSCs. This finding provides a unique perspective on the suppression of nonradiative energy loss in organic photovoltaic technique, which should be of high interest for OSC community. Therefore, I strongly recommend this manuscript to be accepted for publication in Nature communications after addressing the following issues.

Our response: We sincerely thank Reviewer 2 for the very positive comments.

> Compared to inorganic solar cells, larger nonradiative energy loss limits the efficiency promotion of OSCs. Many efforts have been devoted to explore how to reduce nonradiative energy loss and then achieve higher performance. In contrast to early studies, the authors have got a very impressive new finding, and I suggest them to more clearly express the difference and significance of this finding.

Our response: Thanks for your good suggestion.

Electron-phonon coupling is a fundamental and critical issue serving as a restrictive factor for the performance of organic optoelectronic devices when considering the molecular (quasi-)crystalline nature of organic semiconductors. Specifically, electron-phonon coupling plays a more critical role in OSC performance than in traditional inorganic solar cells, e.g., single-crystal silicon solar cells.

Flexible alky side chains are normally utilized to endow organic semiconductors with solution processability and are considered to affect crystallinity but have no influence on electron-phonon coupling. In our manuscript, for the first time, we discovered and confirmed that the rational side-chain engineering can effectively restrict molecular motions to reduce the electron-phonon coupling via intermolecular interactions, and FVR is proposed to describe the electron-phonon coupling in the aggregation state that is otherwise hardly obtained, which provides an effective approach to reduce ΔE_{nr} via rational molecular design. The Chinese idiom “一 瓶子不响，半瓶子晃荡 (The half-filled bottle sloshes; the full bottle remains still.)” can well express our finding. As shown below (**Figure R1**), because of the varied filling extent (free volume ratio) in bottle, the identical water molecules (conjugated backbone) generate different vibration frequency (electron-phonon coupling).

Figure R1. Schematic diagram.

To better express the uniqueness of our work, the following modification in abstract has been made.

“Our study discovered aggregation state regulation is of great importance to the reduction of electron-phonon coupling, which paves the way to high efficiency OSCs.”

> In generally, charge transport property is closely related to the aggregation behavior in solid state, and 3D packing is in favors to form 3D charge transport channels (*Nat. Energy* **2021**, *6*, 605.). As the authors described in the manuscript, AQx-type acceptors possess 2D packing in single crystal, and previous study demonstrated that Y6 shows 3D packing. However, D18:AQx-6 delivers higher performance than that of D18:Y6. Can the authors give more explanations on it?

Our response: Thanks for your valuable comment.

Firstly, the GIWAXS results show that both in AQx-type acceptor neat and blend films possess a favorable face-on orientation, which means that the AQx-type acceptors with 2D packing also can efficiently transport charge carriers to the electrode in the vertical direction. Moreover, D18:AQx-6-based device shows higher electron and hole mobilities of 4.75×10^{-4} and $3.16 \times 10^{-4} \text{ cm}^2 \text{ V}^{-1} \text{ s}^{-1}$ than those in D18:Y-based device (1.40×10^{-4} and $1.49 \times 10^{-4} \text{ cm}^2 \text{ V}^{-1} \text{ s}^{-1}$) (*Sci. Bull.* **2020**, *65*, 272), which allows a high rate of charge transport with less charge recombination. Secondly, the blend films in OSCs consists of abundant amorphous domains and crystal domains with orderly stacking. Generally, the size of crystal domain is limited, but the amorphous packing can perform connection function to form multidimensional charge transport channels between crystal domains, inducing the formation of continuous interpenetrating networks for efficient charge transport. Besides, Wei *et al.* reported Qx-typed acceptors that possesses 1D packing model (*Nat. Commun.* **2022**, *13*, 3256), delivering the high PCE over 18% when blending with PM6, which can further prove that acceptors with 1D or 2D packing model can form multidimensional charge transport channels to transport charge with a high mobility.

> By precisely extending alkyl chain length, AQx-6 can simultaneously exhibit suppressed electron-phonon coupling for low energy loss, high charge transport property and efficient charge generation efficiency. If possible, the performance of AQx-6-based cells should be verified by a trusted third authority.

Our response: Thanks for the reviewer’s good suggestion.

We have provided a certified PCE of D18:AQx-6-based devices from the National Institute of Metrology (NIM), China, and the results are shown in **Figure S13**. The related data were added to the revised manuscript on page 6, as shown below:

“Besides, the certification of D18:AQx-6 performance was carried out in National Institute of Metrology (NIM), China, and a certified PCE of 18.4% with a V_{OC} of 0.874 V, a J_{SC} of 26.7 mA cm⁻² and an FF of 79.1% was obtained (**Figure S13**).”

Figure S13. The certified PCE of D18:AQx-6-based device obtained from National Institute of Metrology (NIM), China.

> In the part of morphology characterization, the experimental results suggested that there is a great difference among three devices. As the authors mentioned in the manuscript, D18:AQx-2-, D18:AQx-6- and D18:AQx-8-based blends show different phase separation behaviors. Considering the alkyl side-chain will affect the material solubility, the miscibility between donor and acceptors should be studied.

Our response: Thank you for the professional advice.

We have provided the miscibility between the donor and acceptors by the contact angle measurements, in which deionized water and diiodomethane (DIM) were selected as the calibration liquids. The film surface free energy (γ) including the London dispersion (γ_d) and polar component (γ_p) were calculated by using the measured contact angles and the corresponding parameters are listed in **Table S9**. The miscibility between the donor and acceptors can be evaluated by the Flory-Huggins interaction parameter (χ), in which χ is defined as $\chi = K(\sqrt{\gamma_D} - \sqrt{\gamma_A})^2$. The relevant discussion was added on page 9 in the revised manuscript.

“The alteration of alkyl side chain will affect the miscibility between D18 and AQx-type acceptors, making difference in the formation of phase separation. The contact angle measurements of neat donor and acceptor films were performed, the corresponding results are shown in **Figure S21** and the relevant data are summarized in **Table S9**. With the increase in alkyl side-chain length, AQx-6 and AQx-8 exhibit reduced surface energies (γ) of 40.35 and 40.61 mN m⁻¹ compared to AQx-2 (42.00 mN m⁻¹). The miscibility between donor and acceptor can be evaluated by the Flory–Huggins interaction parameter (χ), in which χ is defined as $\chi = K(\sqrt{\gamma_D} - \sqrt{\gamma_A})^2$. The χ values are 0.11K for D18:AQx-2, 0.04K for D18:AQx-6 and 0.05K for D18:AQx-8. The lower χ values in D18:AQx-6 and D18:AQx-8 imply favorable miscibility between D18 and AQx-6/AQx-8 with prolonged alkyl chain length, which is beneficial for the formation of desired phase separation.”

Figure S21. The contact angle for D18, AQx-2, AQx-6 and AQx-8.

Table S9. Surface energy of the D18, AQx-2, AQx-6 and AQx-8 pristine films.

Materials	θ_{D18} [deg]	θ_{water} [deg]	γ [mN m ⁻¹]	γ^{p} [mN m ⁻¹]	γ^{d} [mN m ⁻¹]	$\chi^{\text{D-A}}$
D18	50.07	101.68	37.81	0.36	37.56	
AQx-2	35.53	86.56	42.00	0.323	41.681	0.11K
AQx-6	40.86	94.06	40.35	0.004	40.34	0.04K
AQx-8	41.37	96.24	40.61	0.082	40.53	0.05K

> Since the AQx-6 and AQx-8 were firstly reported in this work, the full name of AQx-6 and AQx-8 should be provided.

Our response: We have provided the full name of AQx-2 and AQx-8, which was added in the manuscript in page 3, as shown below:

“Therefore, AQx-2 with symmetrical 2-ethylhexyl, 2,2'-((2Z,2'Z)-((13-(2-ethylhexyl)-14-(2-hexyldecyl)-3,10-diundecyl-13,14-dihydrothieno[2'',3'':4',5']thieno[2',3':4,5]pyrrolo-[3,2-f]thieno-[2'',3'':4',5']thieno[2',3':4,5]pyrrolo-[2,3-h]quinoxaline-2,11-diyl)bis(methaneylylidene))bis(5,6-difluoro-3-oxo-2,3-dihydro-1*H*-indene-2,1-diylidene))dimalononitrile (AQx-6) with asymmetrical 2-ethylhexyl and 2-hexyldecyl and 2,2'-((2Z,2'Z)-((13,14-bis(2-hexyldecyl)-3,10-diundecyl-13,14-dihydrothieno[2'',3'':4',5']thieno-[2',3':4,5]pyrrolo[3,2-f]thieno[2'',3'':4',5']thieno[2',3':4,5]pyrrolo[2,3-h]quinoxaline-2,11-diyl)bis(methaneylylidene))bis(5,6-difluoro-3-oxo-2,3-dihydro-1*H*-indene-2,1-diylidene))dimalononitrile (AQx-8) with symmetrical 2-hexyldecyl were designed and synthesized, with regulated alkyl side chains attached on the nitrogen atoms of the pyrrole rings.”

> The thermogravimetric analysis (TGA) measurements of three acceptors should be performed to study the material thermostability.

Our response: We have performed the TGA measurements (**Figure R10**) of AQx-2, AQx-6 and AQx-8 to analyze the material thermostability, and the relevant discussion was added into the revised manuscript on page 6.

“Moreover, as shown in **Figure S11**, AQx-type acceptors possess good thermal stability (5% weight loss at 345, 347 and 345 °C for AQx-2, AQx-6 and AQx-8, respectively) according to the thermogravimetric analysis (TGA).”

Figure S11. The thermogravimetric analysis (TGA) of AQx-2, AQx-6 and AQx-8.

> While the device optimization process of D18:AQx-6 was summarized in Supplementary information, the author did not give the optimization process of D18:AQx-2 and D18:AQx-8. The related data also need to be provided.

Our response: Thanks for Reviewer 2's valuable suggestion.

We have added the detailed optimization process of D18:AQx-2 and D18:AQx-8 into the Supplementary information (**Table S4-7**).

Table S4. Photovoltaic parameters of OSCs based on D18:AQx-2 treated with different thermal annealing.

Treatment	V_{oc} [V]	J_{sc} [mA/cm ²]	FF [%]	PCE [%]
As cast	0.873	24.5	75.3	16.1
60°C	0.871	25.4	75.3	16.7
80°C	0.868	26.1	76.0	17.2
100°C	0.865	25.9	75.6	16.9
120°C	0.860	25.8	75.5	16.8

Table S5. Photovoltaic parameters of OSCs based on D18:AQx-2 with different D/A ratio at 80°C thermal annealing.

D:A	V_{oc} [V]	J_{sc} [mA/cm ²]	FF [%]	PCE [%]
1:1	0.870	25.3	76.3	16.8
1:1.2	0.868	26.1	76.0	17.2
1:1.4	0.868	25.9	75.3	16.9

Table S6. Photovoltaic parameters of OSCs based on D18:AQx-8 treated with different thermal annealing.

Treatment	V_{oc} [V]	J_{sc} [mA/cm ²]	FF [%]	PCE [%]
As cast	0.914	24.3	75.0	16.7
60°C	0.914	24.9	74.4	17.0
80°C	0.913	24.7	75.7	17.1
100°C	0.909	24.8	74.8	16.9
120°C	0.902	24.9	74.0	16.6

Table S7. Photovoltaic parameters of OSCs based on D18:AQx-8 with different D/A ratio at 80°C thermal annealing.

D:A	V_{oc} [V]	J_{sc} [mA/cm ²]	FF [%]	PCE [%]
1:1	0.914	24.7	74.4	16.8
1:1.2	0.913	24.7	75.7	17.1
1:1.4	0.909	24.9	74.5	16.8

> Due to the weaker π - π stacking in AQx-8 than AQx-2 and AQx-6, AQx-8 presented inferior electron mobility, causing more severe bimolecular recombination, which was verified by the J_{SC} - P_{light} relationship. The recombination mechanism should be discussed in the main text.

Our response: Thanks for Reviewer 2's pertinent comment.

We have investigated the relationship between V_{OC} and light intensity to analyze the dominated charge recombination in AQx-2-, AQx-6- and AQx-8-based devices. The V_{OC} - P_{light} plot is shown in **Figure S15** and the relevant discussion is provided in the manuscript (page 6).

“In addition, charge recombination behavior in the three OSCs was examined by the dependence of V_{OC} and J_{SC} on light density (P_{light}). The relationship between V_{OC} and P_{light} can be described as $V_{OC} \propto (nkT/q)\ln P_{light}$, where k , T , q are the Boltzmann constant, absolute temperature and elemental charge, respectively. When n is close to 1, the bimolecular recombination is the dominated mechanism in devices. As displayed in **Figure S15**, the slope values for AQx-2-, AQx-6- and AQx-8-based devices are 1.05, 1.01 and 1.09 kT/q , respectively, suggesting all devices are subject to bimolecular recombination. The lowest slope value in AQx-6-based OSC means the trap-assisted recombination was well-mitigated.”

Figure S15. The V_{OC} versus light intensity in AQx-2-, AQx-6- and AQx-8-based devices.

Response to Reviewer 3:

> I am leaning towards rejection, and suggest that the revised manuscript be submitted to a more specialized journal.

Our response: We sincerely thank Reviewer 3's for her/his careful review and pertinent suggestions, according to which we carefully revised our manuscript. Besides, the ambiguous issues have also been clarified.

> The 18% PCE is impressive but it is not a record.

Our response: We had tallied up the reported PCE values of binary OSCs and the PCE of 18.6% (certified PCE:18.4%) for D18:AQx-6-based OSCs is the highest PCE (**Figure S12**) when the manuscript was firstly submitted in **December, 2022**. Besides, in this work, we focus to investigate the influence of alkyl side-chain on electron-phonon coupling, demonstrating that the increased alkyl chain length contributes to reduced free volume ratio (FVR) in aggregation state and restrict molecular motion, which is beneficial to low ΔE_{nr} and high performance for OSCs.

Figure S12. Statistics on recently reported bulk-heterojunction binary OSCs.

> Side chain engineering to improve efficiency is not new. Improving efficiency using asymmetric acceptors to tune balance between charge transfer and recombination has already been done before.

Our response: Thanks for the reviewer's comment.

Side chain engineering is of great significance to OSC performance. Many researches were conducted to investigate the side-chain effect on the photovoltaic performance. The alkyl side-chain modulation is an effective approach to regulate the nanomorphology and then optimize the exciton dissociation and charge transport, which is critical to achieve highly performed photon-to-electron conversion. Many efforts were put on ITIC-type acceptors (*J. Am. Chem. Soc.* **2016**, *138*, 2973; *J. Am. Chem. Soc.* **2016**, *138*, 15011; *Nat. Commun.* **2020**, *11*, 6005) and Y6-type acceptors (*Joule* **2019**, *3*, 3020; *Adv. Mater.* **2020**, *32*, 1908373; *Adv. Mater.* **2020**, *32*, 1908205; *Nat. Energy* **2021**, *6*, 605; *Energy Environ. Sci.* **2021**, *14*, 3469; *Energy Environ. Sci.* **2022**, *15*, 2011). Such a phenomenon indicates that the side chain engineering deserves continuing and in-depth study. Asymmetric design of NFAs can optimize the trade-off between charge transfer and recombination (*Nat. Commun.* **2022**, *13*, 2598), leading to an increase in OSCs performance. Such an asymmetric strategy mainly focused on the π -conjugated backbones.

Flexible alkyl side chains are normally utilized to endow organic semiconductors with solution processability and are considered to affect crystallinity but have no influence on electron-phonon coupling. In our manuscript, for the first time, we discovered and confirmed that the rational side-chain engineering can effectively restrict molecular motions to reduce the electron-phonon coupling via intermolecular interactions, and FVR is proposed to describe the electron-phonon coupling in the aggregation state that is otherwise hardly obtained, which provides an effective approach to reduce ΔE_{nr} via rational molecular design. The Chinese idiom “一 瓶子不响，半瓶子晃荡 (The half-filled bottle sloshes; the full bottle remains still.)” can well express our finding. As shown below (**Figure R1**), because of the varied filling extent (free volume ratio) in bottle, the identical water molecules (conjugated backbone) generate different vibration frequency (electron-phonon coupling). Besides, there is an elegant trade-off between the reduced FVR (to suppress electron-photon coupling and achieve low ΔE_{nr}) and the formation of ideal morphology (to form suitable phase separation with favorable π - π packing), which is well addressed by the precise asymmetric side-chain modulation, leading to the best performance in AQx-6-based OSCs. In conclusion, we demonstrate the alkyl chain alteration can help to reduce FVR and allow suppressed electron-phonon coupling, and find that the asymmetric alkyl chain can balance the trade-off between low FVR and ideal morphology, which is different from others research.

Figure R1. Schematic diagram.

To better express the uniqueness of our work, the following modification in abstract has been made.

“Our study discovered aggregation state regulation is of great importance to the reduction of electron-phonon coupling, which paves the way to high efficiency OSCs.”

> I struggled a bit to get the central message of this manuscript by Yuanyuan Jiang *et al.* The arguments need to be better articulated. They are connecting a lower free volume ratio (FVR) with lower delta (E_{nr}) and a higher V_{oc} . The connection seems to be non-linear and they have not discussed other competing processes such as triplet exciton formation for non-radiative energy losses.

Our response: Thanks for Reviewer 3’s pertinent comments. We appreciate Reviewer 3 for her/his insightful comments, allowing us to confirm our discovery with more thorough analysis.

In the section of “Molecular design and theoretical calculations of AQx-type acceptors”, we calculated the FVR of AQx-2, AQx-6 and AQx-8 in aggregation state by theoretical simulation. The FVR values for AQx-2, AQx-6 and AQx-8 are 38.35%, 37.87% and 37.95%, respectively, implying the increased alkyl side-chain length can act as “glue” to reduce FVR. The low FVR indicates suppressed free space for molecular motion in aggregation state, which suggests reduced electron-phonon coupling and minimized nonradiative decay, which is consistent with the experimental data. By using the equation of $PLQY = \frac{k_r}{k_r+k_{nr}} = \tau k_r$, in which the PLQY and τ are photoluminescence quantum yield and carrier life, we can evaluate the radiative decay rate (k_r) and nonradiative decay rate (k_{nr}) of AQx-type acceptors, and the results show that AQx-6 and AQx-8 with prolonged alkyl chain length have lower k_{nr} . Compared with AQx-6, AQx-8 with the longer alkyl chain provides the larger k_r value, leading to a higher PLQY. The ΔE_{LE-CT} values for AQx-2-, AQx-6- and AQx-8-based blends were calculated to be 0.033, 0.031 and 0.034 eV, which indicates the a small and equally strong LE-CT hybridization for AQx-based blend. Considering the LE-CT hybridization, the high emission efficiency of acceptors can lower ΔE_{nr} via an intensity-borrowing process (*Nat. Energy* **2021**, *6*, 799). Therefore, the gradually increased PLQY value from AQx-2 (0.39%) to AQx-6 (0.67%) and then to AQx-8 (1.54%) is beneficial for reduced ΔE_{nr} and higher V_{oc} . The photovoltaic performance and energy loss analysis can further confirm our speculation. OSCs based on AQx-6 and AQx-8 have higher V_{oc} values of 0.892 and 0.913 V and lower ΔE_{nr} of 0.198 and 0.178 eV, respectively, compared to those in OSCs based on AQx-2 (0.868 V and 0.206 eV).

As pointed out by Reviewer 3, the formation of triplet exciton is irreversible, which causes additional nonradiative recombination and thus is unfavorable to reduce ΔE_{nr} in OSCs. Therefore, Transient Absorption (TA) measurements in near-infrared region were performed to study the formation of triplet exciton. TA results (**Figure S20**) show that there are merely small amounts of triplet exciton generated in AQx-based blends, indicating that the formation of triplet exciton is not a dominant factor determining the ΔE_{nr} in OSCs based on AQx-type acceptors. These TA results were double-checked in the labs of Prof. Wenkai, Zhang and Chunfeng, Zhang, independently. We have added the corresponding discussion on the Page 8 in the revised manuscript.

“As exhibited in **Figure 3d**, AQx-2-, AQx-6- and AQx-8-based devices exhibit $E_{QE_{EL}}$ of 3.29×10^{-4} , 4.05×10^{-4} and 8.65×10^{-4} , respectively, corresponding to ΔE_{nr} values of 0.206, 0.198 and 0.178 eV, which is excellently in agreement with the assumption based on theoretical calculation. Considering that the ΔE_{nr} may also be affected by the formation of triplet exciton (T_1), transient absorption (TA) in the near-infrared region was carried out to assess the T_1 formation in AQx-based systems. In the AQx-based blend films, an excited-state absorption (ESA) at ~ 1450 nm emerged after 100 ps (**Figure S20**), which can be ascribed to the T_1 state of the AQx-type acceptor. The ESA of T_1 state emerges less pronouncedly in three AQx-based blend films, suggesting the ΔE_{nr} in AQx systems is not primarily caused by the formation of T_1 state but rather controlled by the effect of electron-photon coupling. From AQx-2 to AQx-6 and AQx-8, the increased alkyl side chain length produces decreased FVR and restricted molecular motion with slow k_{nr} , which indicates a reduction in electron-photon coupling strength and then allows reduced ΔE_{nr} in OSCs. The diminution of ΔE_{nr} greatly promotes the improvement of V_{OC} , thus causing gradually increased V_{OC} from AQx-2 to AQx-6, and then to AQx-8.”

Figure S20. TA profiles (a), TA spectra at different time delay (b) and TA traces of T_1 signals probed at 1450 nm and DSE signals probed at 1550 nm (c) for D18:AQx-2, D18:AQx-6 and D18:AQx-8 blend films with pump at 800 nm.

> The authors later discuss about connections between crystallinity to explain mobilities, J_{sc} and FF. However, they have not done any texture analysis nor have they extracted numbers for the relative degree of crystallinity from the GIWAXS data. As a sidenote, the study would be more complete if they also showed GIWAXS data from neat D18. The authors haven't also made any connection between FVR and the crystallinity. Having RSoXS would further help to characterize the morphology and could further explain the variations in device performance.

Our response: Thanks for the Reviewer 3's valuable suggestions.

We have added the GIWAXS data of D18 (**Figure S24**) into the revised manuscript and Supplementary Information to perfectly analysis the crystallinity of the films. The detailed data extracted for GIWAXS pattern were shown in **Table S10-11**. With the increased alkyl-side chain, AQx-type acceptors (AQx-2, AQx-6 and AQx-8) exhibit weakened π - π stacking, which can be confirmed by the increased π - π stacking distance and reduced crystal coherence length (CCL). Such a trend is also observed in other samples (Nat. Energy 2021, 6, 605). The compact π - π stacking and long CCL indicate higher crystallinity, which contributes to improved carrier mobility and thus endows AQx-2 and AQx-6 with reduced charge recombination for high FF and J_{sc} . The relevant discussion was added into the revised manuscript on page 9.

“From AQx-2 to AQx-8 with the lengthened alkyl chain, the π - π stacking peak (010) gradually shifts to the lower q region, with the specific location at 1.742, 1.737 and 1.704 \AA^{-1} , corresponding to the π - π stacking distances of 3.605, 3.615 and 3.685 \AA , respectively, and demonstrating that the crystallinity is disturbed by the long alkyl chain, which has also been observed in previous reports.²² In **Figure S24**, D18 exhibits a clear (010) peak in the OOP direction, suggesting the favorable face-on orientation. Moreover, the (001) diffraction peak at ~ 0.5 \AA in the IP direction can be used to access crystallization feature of D18.³⁵ Such a peak was observed in the D18:AQx-2 and D18:AQx-6 blend films but disappeared in D18:AQx-8 (**Figure 4b**) suggests that D18 can retain a good crystallinity in AQx-2- and AQx-8-based blends, contributing to improved charge transport. For all blend films, the favorable face-on orientation is maintained, which allows the free charge carriers can be efficiently transported to electrodes in the vertical direction. In the OOP direction, the π - π packing distances for AQx-2-, AQx-6- and AQx-8-based blend films are 3.662, 3.666 and 3.664 \AA , with relevant crystal coherence lengths (CCLs) are 25.01, 20.11 and 17.66 \AA , respectively. The longer CCLs in AQx-2- and AQx-6-based films are conducive to improving charge transport and suppressing charge recombination, while the reduced CCL in AQx-8 may induce more charge recombination. The charge transport property of the three devices can be investigated by the space-charge-limited-current (SCLC) measurement.”

Figure S24. GIWAXS 2D patterns and the corresponding line-cuts of GIWAXS patterns for D18 neat film.

Table S10. The peak position and CCL of different peaks o of pristine AQx-2, AQx-6 and AQx-8 films.

Film	Direction	Q (\AA^{-1})	Stacking distance (\AA)	FWHM (\AA^{-1})	CCL (\AA)
AQx-2	IP	0.274	22.920	0.11	51.38
		0.403	15.583	0.16	35.55
	OOP	1.240	5.065	0.33	17.13
		1.520	4.132	1.21	4.67
		1.743	3.605	0.28	20.48
AQx-6	IP	0.368	17.065	0.19	30.39
		1.230	5.105	0.34	16.62
	OOP	1.436	4.373	1.31	4.3
		1.737	3.615	0.29	19.76
AQx-8	IP	0.313	20.064	0.09	63.51
		1.280	4.906	0.47	12.03
	OOP	1.521	4.129	0.785	7.20
		1.704	3.685	0.272	20.779

Table S11. The peak position and CCL of different peaks o of D18:AQx-2, D18:AQx-6 and D18:AQx-8 blend films.

Film	Direction	Q (\AA^{-1})	Stacking Distance (\AA)	FWHM (\AA^{-1})	CCL (\AA)
D18:AQx-2	IP	0.303	20.726	0.08	69.78
		0.518	12.126	0.22	26.29
	OOP	1.250	5.024	0.40	14.13
		1.472	4.266	0.94	6.03
		1.715	3.662	0.23	25.01
D18:AQx-6	IP	0.310	20.258	0.08	75.36
		0.543	11.565	0.14	41.56
	OOP	1.250	5.024	0.40	14.13
		1.450	4.331	1.06	5.35
		1.713	3.666	0.28	20.11
D18:AQx-8	IP	0.341	18.416	0.16	35.33
		1.260	4.984	0.40	14.13
	OOP	1.455	4.316	0.82	6.893
		1.714	3.664	0.32	17.66

According to Reviewer 3's suggestion, the resonant soft X-ray scattering (RSoXS) was performed to investigate the phase separation in the three blends. As shown in **Figure S23**, D18:AQx-2, D18:AQx-6 and D18:AQx-8 possess domain sizes of 30.72, 21.55 and 22.10 nm, respectively, which may be ascribed to the superior miscibility between D18 and AQx-6/AQx-8. Generally, the domain size of 10–20 nm can guarantee sufficient D/A interface area for exciton dissociation. The more suitable phase separation in D18:AQx-6 and D18:AQx-8 can provide more D/A interfaces for efficient exciton dissociation. The relevant discussion was added to the revised manuscript on page 9.

“Moreover, the resonant soft X-ray scattering (RSoXS), transmission electron microscopy (TEM) and grazing-incidence wide angle X-ray scattering (GIWAXS) measurements were also performed to analyze bulk morphology.⁷⁸ RSoXS is used to investigate the phase separation information in AQx-based blend films. As shown in **Figure S23**, D18:AQx-2 gives the largest domain size of 30.72 nm, while D18:AQx-6 and D18:AQx-8 exhibit phase separation with domain sizes of 21.55 and 22.10 nm, respectively, which is close to the optimal domain size of 10–20 nm generally desired for OSCs. Such a result can be further confirmed by the TEM measurements (**Figure S22**).”

Figure S23. RSoXS profiles of D18:AQx-2, D18:AQx-6 and D18:AQx-8 blend films.

Reviewers' Comments

Reviewer #1 (Remarks to the Author):

The authors have well addressed the reviewers' concerns. The revised manuscript may be publishable in its present form.

Reviewer #2 (Remarks to the Author):

My concerns have been well addressed and I therefore recommend to accept it in its current form.

Reviewer #3 (Remarks to the Author):

The authors have now addressed all my concerns and the manuscript should be accepted in *Nature Communication*.